# Unveiling the *Plasmodium* inositol (pyro) phosphate pathway: Highlighting inositol polyphosphate multikinase as a novel therapeutic target for malaria

Abigail Obuobi[1,2], Neils B. Quashie[1,2,3], Nancy Odurowah Duah-Quashie[2], Jon R. Sayers[4,5]*

**1** Department of Biochemistry, Cell and Molecular Biology, University of Ghana, Accra, Ghana, **2** Department of Epidemiology, Noguchi Memorial Institute for Medical Research, University of Ghana, Accra, Ghana, **3** Centre for Tropical, Clinical Pharmacology and Therapeutics, University of Ghana Medical School, Accra, Ghana, **4** Clinical Medicine, School of Medicine and Population Health, University of Sheffield, Sheffield, United Kingdom, **5** Florey Institute of Infection, University of Sheffield, Sheffield, United Kingdom

* j.r.sayers@sheffield.ac.uk, jon.sayers@yahoo.co.uk

## Abstract

*Plasmodium falciparum* malaria is fatal if left untreated. Treatment is hampered by drug-resistant variants of the malaria parasite, highlighting the need to explore unique pathways for the development of new drugs with different mechanisms of action. Kinases in the inositol phosphate signaling pathway (IPP), and its products play many important roles in energy metabolism and signal transduction, making them attractive drug targets. In this exploratory study we investigated the potential of *P. falciparum* IPP as a novel and attractive pathway for antimalarial drug discovery, employing a combined *in silico* and molecular approach. The sequences and structures of the putative *P. falciparum* inositol phosphate kinases were characterized *in silico*. Experimental validation across laboratory strains and a clinical isolate confirmed the p.Pro375Gln substitution in IPMK1, providing the first evidence of this variant in field isolates. We provide molecular evidence of the existence of IPP genes in *P. falciparum* and suggest that targeting this pathway could be detrimental to the parasite. We identify *P. falciparum* inositol polyphosphate multikinase (IPMK) as a promising drug target due to its unique sequence and structural characteristics. These results serve as a guide for future experimental validation.

## Introduction

Malaria is a tropical infectious disease which is still a global health challenge. Malaria cases are actively rising with a global incidence of 263 million cases and 597,000 deaths in 2023: an increase of 11 million cases from the previous year. The WHO

**Data availability statement:** All relevant data are within the manuscript and its Supporting Information files.

**Funding:** AO : Erasmus+ programme of the European Union Grant: 2020-1-UK01-KA107-078009) https://erasmus-plus.ec.europa.eu The funder played no role in the study design, data collection and analysis, decision to publish, or preparation of the manuscript.

**Competing interests:** The authors have declared that no competing interests exist.

African Region recorded 94% of these cases and 95% of the reported deaths [1]. The emergence of drug-resistant strains, which threaten current treatment plans, adds to this worrying burden on the economy and health, necessitating the urgent identification of new therapeutic targets [2–5]. This underlines the urgent need to investigate hitherto underexplored biochemical pathways in malaria parasites as potential targets for drug development, particularly those with alternative mechanisms of action to those of the existing antimalarials.

Understanding parasite biochemistry is crucial for blocking transmission and tackling drug resistance [6–8]. *Plasmodium falciparum*, the most virulent of all human-infective malaria parasites, cycle between mosquitos and humans. The complexity of studying the parasite stems from the fact that its survival is controlled by the differential expression of genes across these stages in two hosts, although some genes are expressed throughout the parasite's life. Key players in DNA transcription, translation and protein synthesis, as well as signalling and metabolism, are desirable targets in drug discovery [9].

In *Plasmodium* sp., protective immunity is achieved through synchronous development in the erythrocytic stages, an event linked to inositol 1,4,5-trisphosphate (IP3)-mediated calcium release from the intracellular stores in the endoplasmic reticulum [10–12]. Calcium ions are well-known secondary messengers whose signalling is highly conserved in eukaryotes. The mechanisms of calcium homeostasis in multicellular organisms such as humans have been thoroughly investigated, but our understanding of calcium signalling and its homeostasis in parasites, particularly the apicomplexan parasite *P. falciparum,* remains incomplete; especially when, to the best of our knowledge, no IP3 receptor has been identified to date in *Plasmodium* [12,13].

Soluble inositol phosphate kinases (IPKs) are members of the inositol (pyro)phosphate pathway (IPP) involved in calcium and phosphate homeostasis. IPKs, together with membrane-bound phosphoinositide kinases (PIKs), work in concert to orchestrate a repertoire of cellular processes in response to external stimuli in eukaryotes [14]. There are seven PIPKs in the *P. falciparum* genome, and this group of kinases has attracted considerable attention in recent years, mainly because of the success of *P. falciparum* phosphatidylinositol 4-kinase type III beta (PI4KIIIβ) as a clinical drug target [15]. To date, MMV390048, a *P. falciparum* PI4KIIIβ inhibitor, has advanced to phase II clinical trials but faces challenges such as poor solubility and embryofetal toxicity that need to be addressed [16–18]. These challenges underline the need to expand the target scope beyond traditional phosphoinositide kinases to include less well-characterised enzymes such as IPKs.

Knowledge of *P. falciparum* IPKs is largely based on bioinformatics studies where Laha, Portela-Torres [19] mentioned the presence of two members of inositol polyphosphate kinase superfamily and one diphosphoinositol pentakisphosphate kinase in *P. falciparum.* In eukaryotes, soluble inositol phosphates (IPs) are produced by four classes of distinct kinase families: inositol polyphosphate kinase, inositol pentakisphosphate 2-kinase (IPPK), inositol 1,3,4-trisphosphate 5/6-kinase (ITPK1) and diphosphoinositol pentakisphosphate kinase (PPIP5K), also known as VIP [19–23].

Despite extensive research into *P. falciparum* kinase families, prior studies have failed to address kinases in the inositol (pyro)phosphate pathway, which also have many of the conserved structural features of protein kinases. Enzymes involved in inositol phosphate signalling are attracting considerable interest because of the functions of their products in cells and the other noncatalytic activities of some members of the group. Together, these IPKs metabolize IP3 produced by the hydrolysis of phosphatidylinositol 4,5-bisphosphate (PIP2) via phospholipase C hydrolysis to inositol pyrophosphates (PP-IPs) [10,19]. Inositol pyrophosphates are important high-energy moieties involved in phosphate sensing, cell cycle regulation and signalling [24,25]. These kinases and their high-energy products in humans, yeast and kinetoplasts, *Trypanosoma brucei* and *T. cruzi,* have been extensively studied and implicated in a wide array of metabolic activities and nuclear processes [24,26–33].

The existence of the inositol (pyro)phosphate pathway in *P. falciparum* remains elusive. IPMK might be essential for *P. falciparum* because it's a multifunctional enzyme that is involved in calcium homeostasis, chromatin remodelling and metabolic adaptation which may promote survival, immune evasion, and drug resistance. This makes it an appealing candidate for further investigation as potential drug target. Thus, we hereby identify, characterize and molecularly confirm the presence of the specific genes encoding these IPKs in *P. falciparum* and suggest that the *P. falciparum* inositol (pyro) phosphate pathway is a novel antimalarial node.

## Methodology

### Identification and retrieval of sequences

The identities of the four putative genes of the *P. falciparum* IPKs were obtained from PlasmoDB (http://PlasmoDB.org), an official functional database of the genus *Plasmodium* [34]. The genes were searched using the keywords '3D7 + their names', after which sequences of 3D7 annotation were selected. The initial data obtained here included the gene IDs, sequences, potential essentiality, protein expression information and functional prediction of the kinases, which supported our selection of the identified genes as *P. falciparum* IPKs.

### Phylogenetic analysis and comparative genomics

To search for conserved domains and motifs, the full-length *P. falciparum* IPK protein sequences were subjected to an NCBI Conserved domain search https://www.ncbi.nlm.nih.gov/Structure/cdd/wrpsb.cgi [35] and the UniProtKB search tool https://www.uniprot.org/ [36]. Once conserved domains were identified, the kinase domains were extracted and used for all subsequent comparative, phylogenetic, and structural analyses. The kinase domain sequences of *P. falciparum* IPKs were also aligned with those of other organisms via the PROMALS3D webserver http://prodata.swmed.edu/ [37] and visualized and analysed using Jalview software [38–40]. The percentage identities were calculated in JalView by pairwise alignment of the kinase domain sequences of interest. In addition, BLASTp searches [41] were performed using the kinase domain sequences against the complete *Homo sapiens* proteome filtered with an "expect value" E value $\leq 10^{-5}$, percentage identity < 35% and a query coverage of > 60% was performed to identify potential orthologues in humans.

The evolutionary relationships between the aligned *P. falciparum* IPK domains and those of the selected organisms were investigated via the maximum likelihood method via the IQ-TREE web server http://iqtree.cibiv.univie.ac.at/ [42]. The PROMALS3D alignment of the amino acid sequences corresponding to the IPK domains was used to search for the best substitution models for the analysis. The Ultrafast bootstrap test method [43] with 1000 replicates was used for both analyses, and the trees generated were visualized and analysed using the online tool iTOL v6.0 https://itol.embl.de/ [44].

### Prediction of protein–protein interactions

The *P. falciparum* IPK protein sequences were then submitted to the STRING (Version 12.0) database https://string-db.org/ [45] to elucidate potential protein–protein interactions, including functional, physical or nearby genes.

## Structure prediction and analysis of kinase domains in *P. falciparum* IPKs

The 3D structures of the full-length kinases predicted by AlphaFold were obtained from the UniProtKB database https://www.uniprot.org/ [36,46], and their functional domains were extracted using PyMOL (The PyMOL Molecular Graphics System, Version 2.4.1 Schrödinger, LLC). The structures of the kinase domains of the *P. falciparum* IPKs were compared with those of the *H. sapiens* IPKs via superimposition in PyMOL (The PyMOL Molecular Graphics System, Version 3.0 Schrödinger, LLC). This approach was used to determine the sequence-to-structure divergence between the *P. falciparum* IPKs and their human counterparts. Flexible structure alignment was performed by submitting the domain structures to the RCSB PDB portal's pairwise structure alignment tool https://www.rcsb.org/alignment?uuid=6c728477-3b1b-438e-afa7-89060717d1b5, and the jFATCAT flexible algorithm was selected as the alignment method [47,48].

## *Ab initio* modelling of *P. falciparum* IPMK1 and 2 domains for molecular docking

*P. falciparum* IPMKs have no homologues in humans or any other organism with experimentally solved structures; thus, the 3D structure of the kinase domain was predicted from its amino acid sequence. To predict the structure of the *P. falciparum* IPMK1 domain *ab initio*, the kinase domain sequence of *P. falciparum* IPMK1 was submitted to Alphafold2 Colab https://colab.research.google.com/github/sokrypton/ColabFold/blob/main/AlphaFold2.ipynb [46,49]. All default settings were maintained except for the number of cycles, which was set at a maximum of 48 cycles and runs on a T4 GPU for fast predictions. All 5 predicted structures were not Amber relaxed. For 3D structure prediction for *P. falciparum* IPMK2, the amino acid sequence corresponding to its kinase domain was subjected to the SWISS MODEL webserver at https://swiss-model.expasy.org/ [50].

All the predicted 3D structures were refined using the GalaxyRefine web server (http://galaxy.seoklab.org/refine). In addition to improving model quality, GalaxyRefine also performs energy minimization [51]. To analyse the quality of the refined structures of *P. falciparum* IPMKs, the structures were further validated on the SAVES meta server (https://saves.mbi.ucla.edu/) along with PROCHECK [52] to construct a Ramachandra plot.

## Identification of active site residues

One particularly important aspect of structure-based drug discovery is the identification of druggable pockets and active residues. The active residues of *P. falciparum* IPMKs were predicted via comparative analysis of the protein sequence/structure alignment of the domain of *P. falciparum* IPMK1 and 2 with that of PDB:5W2H, which is a crystal structure of *H. sapiens* IPMK in complex with ADP, IP3 and magnesium ions [22,53]. The PROMALS3D web server (http://prodata.swmed.edu/promals3d/) [37] was used to construct alignments for multiple protein domain sequences (MSAs). 3D structural alignment was performed in PyMOL 2.0 (The PyMOL Molecular Graphics System, Version 2.0 Schrödinger, LLC).

## Molecular docking

The structure of ATP in SDF format was downloaded from the RCSB.org [54], energy minimized and converted to a pdbqt file in PyRx and used for the docking work. Site-specific molecular docking for ATP was performed in PyRx with AutoDock Vina software, and the exhaustiveness was set at 32. Grid boxes were centred around residues Lys[9], Glu[364], Val[366], Asp[379] and Asp[624] for *P. falciparum* IPMK1 and Trp[1], Lys[14], Ser[204], Val[206], Asp[217] and Asp[442] for *P. falciparum* IPMK2. The docking results were analysed in PyMOL 2.0 (The PyMOL Molecular Graphics System, Version 2.0 Schrödinger, LLC).

## Druggability assessment

The druggability of *P. falciparum* IPMK2 was assessed using the PockDrug-Server https://pockdrug.rpbs.univ-paris-diderot.fr/cgi-bin/index.py?page=home [55] to assess its potential for inhibition by small compounds. This server is useful for predicting druggability and identifying druggable pockets, which is paramount in target identification and validation. On the

webserver, "predict druggability using proteins" was selected, the AlphaFold-predicted kinase domain 3D structure was uploaded, and the Fpocket algorithm was used. All other parameters remained at their default settings.

## Subcellular localization, functional and analyses

The possible subcellular locations of *P. falciparum* IPKs were investigated by subjecting their protein sequences to the webservers BUSCA (http://busca.biocomp.unibo.it) and Euk-mPLoc2.0 (http://www.csbio.sjtu.edu.cn/bioinf/euk-multi-2/) [56–58]. Predicting subcellular localization is important for inferring functions and understanding potential signaling pathways, as has been extensively reviewed [59]. The potential catalytic functions of the proteins were derived from PlasmoDB, NCBI-protein and UniProtKB. In addition, other potential noncatalytic activities of the IPKs were investigated via the PredictProtein webserver (https://predictprotein.org/).

In addition, stage-specific expression data for the IPK genes were reviewed from mass spectrometry-based proteomic studies deposited in PlasmoDB [60] (referenced in the Results), as transcript-level quantification was not within the scope of this study. These datasets enabled us to infer life stage-specific protein expression without direct validation of mRNA or protein.

## Design of primers

Primers specific to the putative *P. falciparum* IPKs were designed with the NCBI Primer–BLAST tool gene sequences obtained from the NCBI gene database. For gene detection, primers flanking the full-length genes or major regions were designed. The quality of the primer pairs was then assessed with Sequence Manipulation Suite (Version 2) PCR Primer Stats https://www.bioinformatics.org/sms2/pcr_primer_stats.html [61], and *in silico* PCR was performed with *AmplifX* software version 2.2.1. All primer sequences are listed in Table 1.

## Ethics statement

The Institutional Review Board of the Noguchi Memorial Institute for Medical Research reviewed and approved this study (Protocol NMIMR-IRB CPN 028/22–23). At the Kpone Polyclinic in the Kpone-Katamanso Municipal District, Greater Accra Region, Ghana, a 16-year-old outpatient provided one clinical isolate. Before the patient could participate, their legal guardians gave their written informed consent. The approval covered the entire duration of the study and was valid until 8 December 2023.

## Sample collection

Cultured strains of *P. falciparum* 3D7 and Dd2 were obtained from the Laboratory of Professor Neils Quashie at the NMIMR. The parasites were cultured in RPMI-1640 (Thermo Fisher Scientific, Catalogue No. 13430287) medium containing human O+red blood cells at 37 °C in a mixed gas incubator. One clinical isolate was obtained from an outpatient 16 years of diagnosed with uncomplicated malaria at the Kpone polyclinic.

**Table 1. Sequences of Primers used in the amplification of *P. falciparum* IPKs.**

| Gene | Forward primer | Reverse primer |
|------|----------------|----------------|
| IPMK1 | TGAACAACTAGCCATTCCGT | TAAATAGGCTCCCCCATCCC |
| IPMK2 | GGAGAGCCATAATACATCAAACGA | ACGGTATCATTCGTTACATCCG |
| IPK2 | TCAAGTGGGAGGACACTGC | GGGAACTTTCTGAAGAATAGGTGT |
| PPIP5K | TGGAAAGCAAAGTTGAGAGTGC | GGTGTGCCCATAAATCTGAGGA |

## Genomic DNA extraction and PCR amplification

The *P. falciparum* infection was microscopically confirmed by Giemsa-stained thin blood smears prior to DNA extraction. Genomic DNA of the parasites was extracted from 200 µL of whole blood samples from clinical isolate and cultured 3D7 and DD2 strains using a QIAamp DNA Blood Mini Kit (Qiagen, Catalogue No. 51104) according to the manufacturer's protocol and stored at −20 °C until use. Amplification was performed using Eppendorf MasterCycler Nexus Gradient Thermal Cycler.

### *P. falciparum I*PMK1 PCR

The *P. falciparum* IPMK1 gene was amplified by conventional PCR using primer sequences that flank the entire *P. falciparum* IPMK gene, and all amplification reactions were performed in a final reaction volume of 15 µL, with 0.3 µL (0.2 µM) of each primer, 0.3 µL of 10 mM dNTPs (New England Biolabs, Catalogue No. M0267S), 3.0 µL of 5X One*Taq* Standard Reaction Buffer, 0.18 µL of One*Taq* DNA Polymerase (NEB, Catalogue No. M0480S) and 2 µL [53 ng F1; 25 ng 3D7; 21 ng DD2] template DNA. For the cycling conditions, initial denaturation was set at 94 °C for 60 s, followed by 30 cycles of denaturation (94 °C for 30 s), annealing (52.7 °C for 30 s) and extension (68 °C for 2:40 s), with a final extension of 5 min at 68 °C.

### *P. falciparum* IPMK2 PCR

Primers specific to the gene sequences corresponding to the kinase domain of the HDAC2 gene were used to amplify the *P. falciparum* IPMK2 gene. Amplification reactions and cycling conditions were carried out as with *P. falciparum* IPMK1, except that 0.2 µL of each primer and 1 µL [26 ng F1; 13 ng 3D7; 11 ng DD2] template was used, and the annealing temperature was set to 50 °C with 30 cycles of extension at 68 °C for 1:33 s; all other conditions remained the same.

### *P. falciparum* IPK2 PCR

For the detection of *P. falciparum* IPK2 gene, primers specific to the IPK2 gene sequence were used. The reactions were performed to a final volume of 15 µL containing 7.5 µL of OneTaq® 2X Master Mix with Standard Buffer (NEB, Catalogue No. M0486S) supplemented with 0.1 µL of One*Taq* DNA Polymerase (NEB, Catalogue No. M0480S), 0.225 µL (0.15 µM) of each primer and 1 µL [26 ng F1; 13 ng 3D7; 11 ng DD2] template, and the cycling conditions remained the same as those for *P. falciparum* IPMK1.

### *P. falciparum* PPIP5K PCR

The gene sequence representing the kinase domain was amplified in a 15 µL reaction as in the protocol of *P. falciparum* IPMK1, with the exception that 1 µL template was used, and for the thermocycling conditions, the annealing temperature was set to 54.8 °C for 30 s with 30 cycles of extension at 68 °C for 2:30 s.

All reactions included a no-template control (nuclease-free water), which was run in parallel with the sample reactions to rule out contamination in the PCR workflow.

## Agarose gel electrophoresis and sequencing

All PCR products with expected sizes of approximately 3.3 kb, 1.6 kb, 4.8 kb and 4.8 kb for *P. falciparum* IPMK1, IPMK2, IPK2 and PPIP5K, respectively were subjected to 0.8 to 1% agarose gel electrophoresis, stained with SYBR™ Safe DNA gel stain (Invitrogen™, Thermo Fisher Scientific, Catalogue No. S33102) and visualized in an iBright™ CL1500 Imaging System (Invitrogen™, Thermo Fisher Scientific). Amplicon sizes were determined relative to the standard molecular weight marker, a 1 kb DNA ladder (Promega Corporation, Catalogue No. G5711).

Amplicons from full length gene were subjected to next-generation sequencing on the Illumina MiSeq platform using the protocol described by Talundzic, Ravishankar [62].

## Results and discussion

### The identity of *P. falciparum* inositol phosphate kinases (IPKs)

The four *P. falciparum* IPK genes under study were identified in PlasmoDB (http://PlasmoDB.org) [34] as putative *P. falciparum* genes with the IDs PF3D7_0514800 (Inositol polyphosphate multikinase, *IPMK1*), PF3D7_1316100 (Inositol polyphosphate kinase, *IPK2*) and PF3D7_1430300 N_Domain (Diphosphoinositol pentakisphosphate kinase, *PPIP5K*). The fourth gene, with the gene ID PF3D7_1008000 annotated as histone deacetylase 2 (HDA2), is a multidomain gene harbouring *P. falciparum* IPMK2 as a domain; all four genes are intronless.

*P. falciparum IPMK* and *IPK2* are located specifically at Pf3D7_05_v3:615,232..618,202(-) and Pf3D7_13_v3:673,163..678,530(+), respectively. The NCBI conserved domain database [40] revealed domains in these sequences showing that *P. falciparum IPMK1* and *IPK2* indeed encode proteins belonging to the superfamily of inositol phosphokinases, with the predicted catalytic domain located at the N-terminus of *P. falciparum* IPMK1 and the C-terminus of the IPK2 protein.

*P. falciparum IPMK2 and PPIP5K,* on the other hand, are part of multidomain coding genes located at Pf3D7_10_v3:320,042..328,566(+) and Pf3D7_14_v3:1,191,911..1,200,002(-), respectively. *P. falciparum* IPMK2 is part of a multidomain gene containing a histone deacetylase domain, an arginase-like/histone-like hydrolase superfamily domain and the C-terminal IPK domain herein reported as *P. falciparum* IPMK2. As its annotated name (acid phosphatase) suggests, it encodes a polypeptide of 2657 amino acids with a mid-terminal histidine phosphatase domain, *P. falciparum* PPIP5K_N VIP1 N-terminal domain next to a C-terminal glutathione synthetase ATP-binding domain (NCBI Conserved Domain Database).

According to the PlasmoDB and UniProtKB deposited E.C. numbers and InterPro scan data of the putative genes encoding the enzymes of the soluble IP pathway, *P. falciparum* may have two IPMKs, an IP6K/IPK2 and a PPIP5K1 enzyme in the inositol pyrophosphate pathway. The potential essentiality of the *P. falciparum* IPKs was also obtained from the data retrieved from the PlasmoDB database, which was based on the original work of Zhang, [63] and is summarized in Fig 1.

### Identification of conserved motifs and residues

Inositol phosphate kinases are generally categorized into sub-families based on their functional motifs. One group is the "PDKG kinases", named for the presence of the PxxxDxKxG motif, also contain SLL and IDF motifs [64,65]. The motif PxxxDxKxG is the catalytic signature of PDKG kinases. All three motifs are involved in ATP binding, but only PxxxDxKxG and IDF participate in substrate binding [66,67]. Multiple kinase domain sequence alignment generated (Fig 2), revealed the presence of all three motifs in *P. falciparum* IPMK1, IPMK2 and IPK2.

However, in *P. falciparum* IPMK1, a conserved substitution was observed in the canonical PxxxDxKxG motif, where glutamine (Q) replaced proline (P) at the first position, resulting in a QxxxDxKxG variant motif. This variant has not been previously reported or functionally described in *Plasmodium* or other organisms to the best of our knowledge. Notably, this motif was experimentally confirmed through Next-generation sequencing in the field isolate, suggesting that it is present in at least one naturally occurring *P. falciparum* strain. In contrast, *P. falciparum* IPMK2 and IPK2 retained the canonical PxxxDxKxG motif**.**

To determine whether this divergent motif was unique to *P. falciparum* or conserved more broadly across the four other plasmodial species that infect humans, a BLAST search within PlasmoDB was performed, and alignment was analysed using JalView software [38]. In addition to the SLL and IDF motifs, all orthologous IPMK1 sequences from *P. vivax, P. knowlesi, P. ovale, and P. malariae* also contained the QxxxDxKxG motif instead of the canonical PxxxDxKxG motif (Fig S1 in S1 File). This observation suggests that the QxxxDxKxG motif is not *P. falciparum*–specific but rather a conserved distinguishing feature of the *Plasmodium* IPMK1 clade.

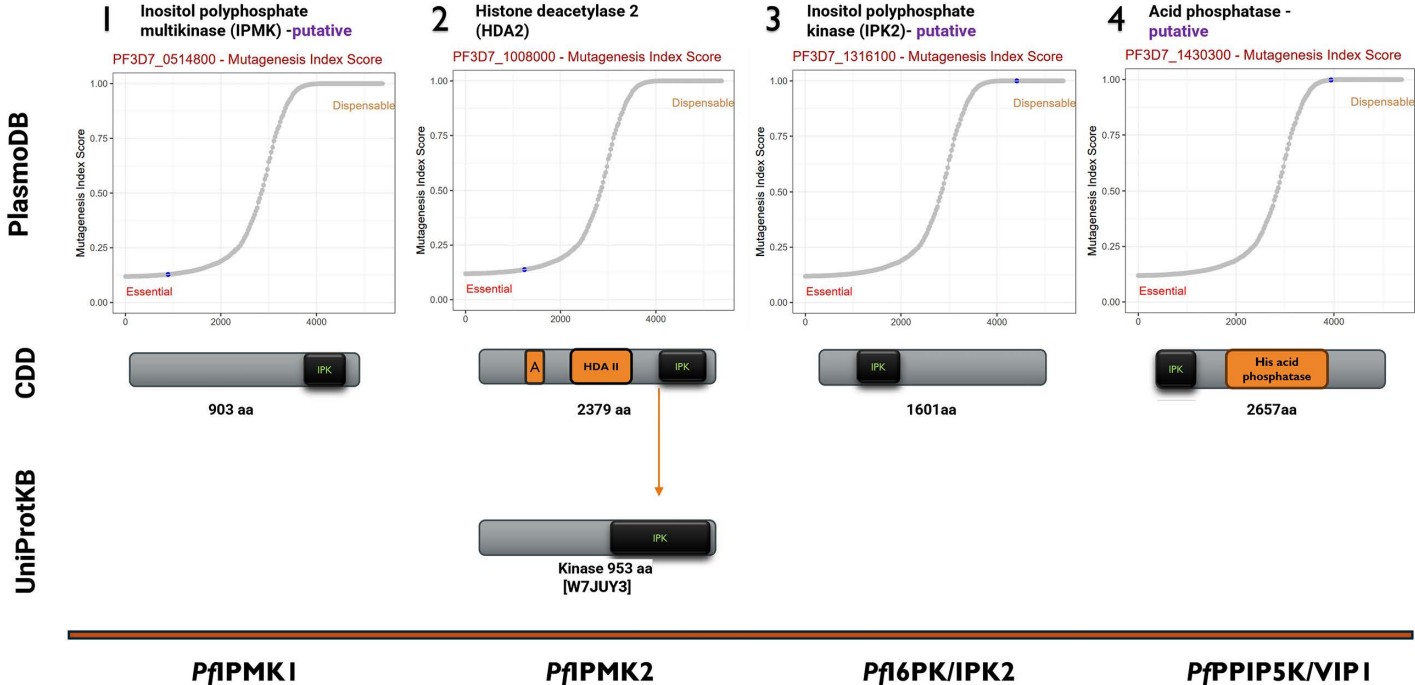

**Fig 1. The identity of the putative *P. falciparum* inositol phosphate kinases.** According to PlasmoDB, both *P. falciparum* IPMKs are essential but not *P. falciparum* IPK2 and PPIP5K. The NCBI CDD search revealed the locations of *P. falciparum* IPKs. *P. falciparum* IPMK2 has been annotated as a standalone protein; UniProtKB annotates a kinase with the ID W7JUY3. The essentiality graph was obtained from the PlasmoDB database [34]. A in PF3D7_100800 is a domain belonging to the arginase-like/histone-like hydrolase (PFAM00850) super family, next to the HDACII domain.

As motif conservation within a group of amino acid sequences from evolutionarily related organisms implies similar characteristic features and functions, it is expected that *P. falciparum* IPMK1 will retain the assigned functions of the SLL and IDF motifs. However, the QxxxDxKxG motif presents a notable substitution in which a chemically distinct amino acid, glutamine (Q), replaces proline (P). This change may significantly alter the structure and function of the protein, particularly affecting ATP binding. Such a substitution is likely to disrupt the conserved interaction network, thereby impacting catalytic activity. While Stritzke, et al [68] noted the absence of the canonical PxxxDxKxG motif in the hypothetical *P. knowlesi* protein XP_002259666.1, our analysis identifies this protein as a syntenic ortholog of *P. falciparum* IPMK1 and confirms that it harbours a conserved QxxxDxKxG motif. This supports the motif as a defining feature of the *Plasmodium* IPMK1 subfamily (Fig S1 in S1 File) rather than a genome annotation artefact, thereby extending the observations of Stritzke, [68].

PPIP5K or VIP enzymes have a dual-domain structure that is highly conserved across most eukaryotic species, including *S. cerevisiae, Saccharomyces pombe, Mus musculus, H. sapiens, Drosophila melanogaster, Caenorhabditis elegans, Dictyostelium discoideum, Arabidopsis thaliana, Oryza sativa, Rhodacmea filosa, Giardia intestinalis, Cryptosporidium parvum, Phytophthora infestans* and *Emiliania huxleyi* [19,28,69,70]. Given the high conservation of PPIP5K enzymes, we analysed multiple sequence alignments (MSA) of the conserved kinase domain regions of VIP1s to predict the catalytic residues of *P. falciparum* PPIP5K with reference to initial work on the *H. sapiens* PPIP5K1 protein (Q6PFW1) [69]. It was evident from the alignment (Fig 3) that there was notable conservation within the kinase-ATP-grasp domain within *P. falciparum* VIP1 and *H. sapiens* PPIP5K1 and we identified D315/Asp[315] as the putative catalytic residue required for the kinase activity of *P. falciparum* VIP1. This aligns with previous reports on the catalytic residue of *H. sapiens* PPIP5K1

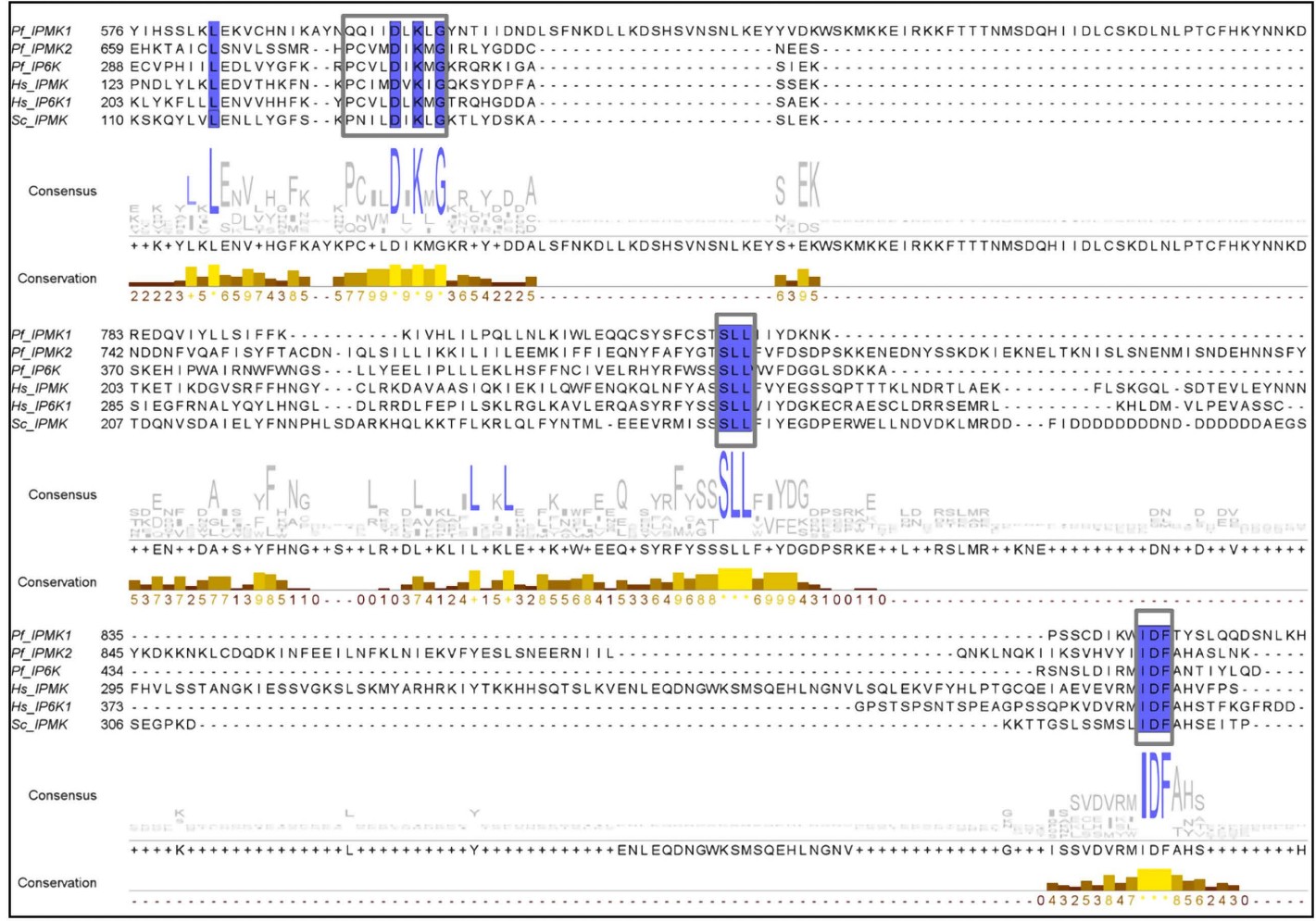

**Fig 2. Multiple sequence alignment of the amino acid sequences of the PDKG kinase domains.** The amino acid sequences of the IPK domains of *P. falciparum* IPMK1 (Q8I3W0), IPMK2 (W7JUY3) and IP6K (Q8IEF3) were aligned with three reviewed sequences in UniProtKB from *H. sapiens* IPMK (Q8NFU5) and IP6K1 (Q92551) and *Saccharomyces cerevisiae* IPMK (P07250). The consensus lane shows the relative distribution of amino acids in each column of the alignment; blue residues are highly conserved residues (shown as a logo), and grey residues are less conserved. The conservation lane is displayed as a quantitative histogram showing the physicochemical properties of each column in the alignment, with scores ranging from a maximum of 11 (as *) or 10 (as +) to a minimum of 0. The blue shaded residues are invariant residues in each column of the alignment. The grey framed box sequences contain the signature motifs identified through the MSA. MSA was performed via the PROMALS3D webserver and viewed in Jalview.

(Asp[321]) and *S. cerevisiae* VIP1 (Asp[485]), suggesting a conserved catalytic mechanism across species as previously reported (Fig 4) [28,71,72]. *P. falciparum* VIP1 also has the RimK/acid-phosphatase dual domain, with the RimK/ATP-grasp domain specially conserved among eukaryotes [72,73].

The full kinase domain in these proteins was thus predicted to be located between the C-terminal residues 401–885 for *P. falciparum* IPMK1, the C-terminal residues 460–951 for *P. falciparum* IPMK2, the N-terminal residues 1–476 for *P. falciparum* IP6K and the N-terminal residues 39–133 with an ATP-grasp domain with residues 185–319 for *P. falciparum* VIP1. The overall architecture of the kinase domains in *P. falciparum* IPKs is shown in Fig 4 below.

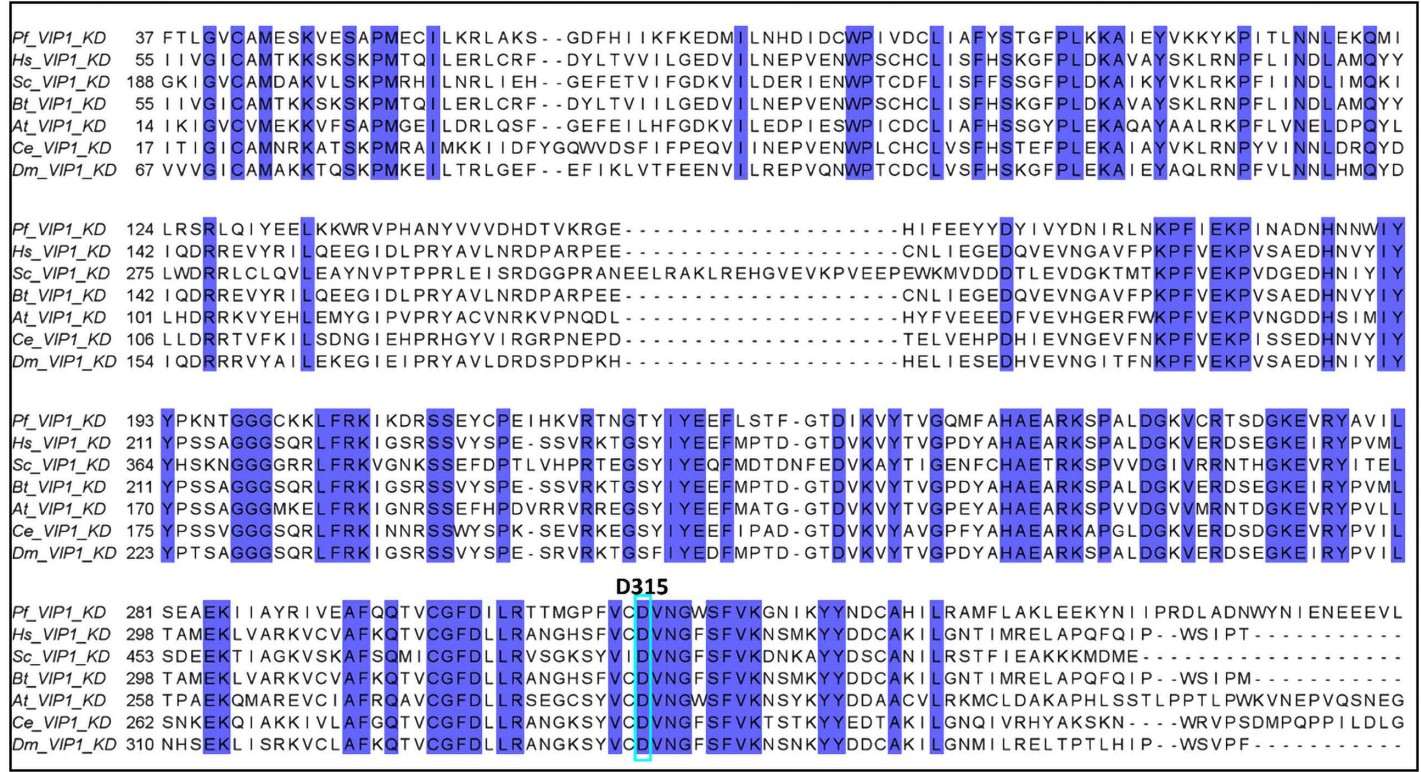

**Fig 3. Multiple amino acid sequence alignment of VIP1 domains.** VIP1 kinase domain sequences from *P. falciparum* (Q8ILG1), *H. sapiens* (Q6PFW1), *D. melanogaster* (Q9VR59), *Bos taurus* (A7Z050), *S. cerevisiae* (Q06685), *A. thaliana* (F4J8C6) and *C. elegans* (P91309) were aligned with PROMALS3D, viewed and analysed in Jalview. Cyan-framed aspartate (D315) is the catalytic residue required for kinase activity. The blue shaded residues are perfectly conserved residues.

## Phylogenetic analysis and comparative genomics

A fast strategy used in identifying potential essential proteins in infectious disease is comparative genomics, which can reveal proteins that are restricted to a given species and non-homologs. When the *P. falciparum* IPK kinase domain sequences were subjected to a BLASTp search against the human proteome database, no significant similarity was found for *P. falciparum* IPMK1, which means that it has no close relationship with any human protein. Although *P. falciparum* IPMK2 and IPK2 shared some sequence similarity with human IPKs, the homology was not significant. The best match obtained for *P. falciparum* IPMK2 and IPK2 was *H. sapiens* inositol hexakisphosphate kinase isoform 3 (IP6K3) (XP_005248900.1). *H. sapiens* IP6K3 matched *P. falciparum* IPMK2 with an E value of $3 \times 10^{-10}$ and a percent identity of 28.5% with a query coverage of 24%, and it matched *P. falciparum* IPK2 with an E value of $1 \times 10^{-34}$ and a percent identity of 39%, which covered 38.94% of the query sequence. These levels of similarity are considered low and unlikely to support functional homology. In contrast to the IPMK proteins, *P. falciparum* VIP1 showed higher homology to its human counterpart, *H. sapiens* PPIP5K1 (XP_047289346.1). The alignment had an E value of $1 \times 10^{-102}$, a high query coverage of 98% and a moderately high sequence identity of 44.91%.

The pairwise alignment of the kinase domains with Jalview showed that the putative *P. falciparum* IPKM1 and IPMK2 share a sequence identity of 22% and 20% respectively with *H. sapiens* IPMK, whereas *P. falciparum* IP6K proteins have 21% and 25% identity with *H. sapiens* IPMK and IP6K1 proteins, respectively. However, *P. falciparum* VIP1, showed a relatively high sequence identity of 45% to the *H. sapiens* VIP1 kinase domain, which agrees with the BLASTp output.

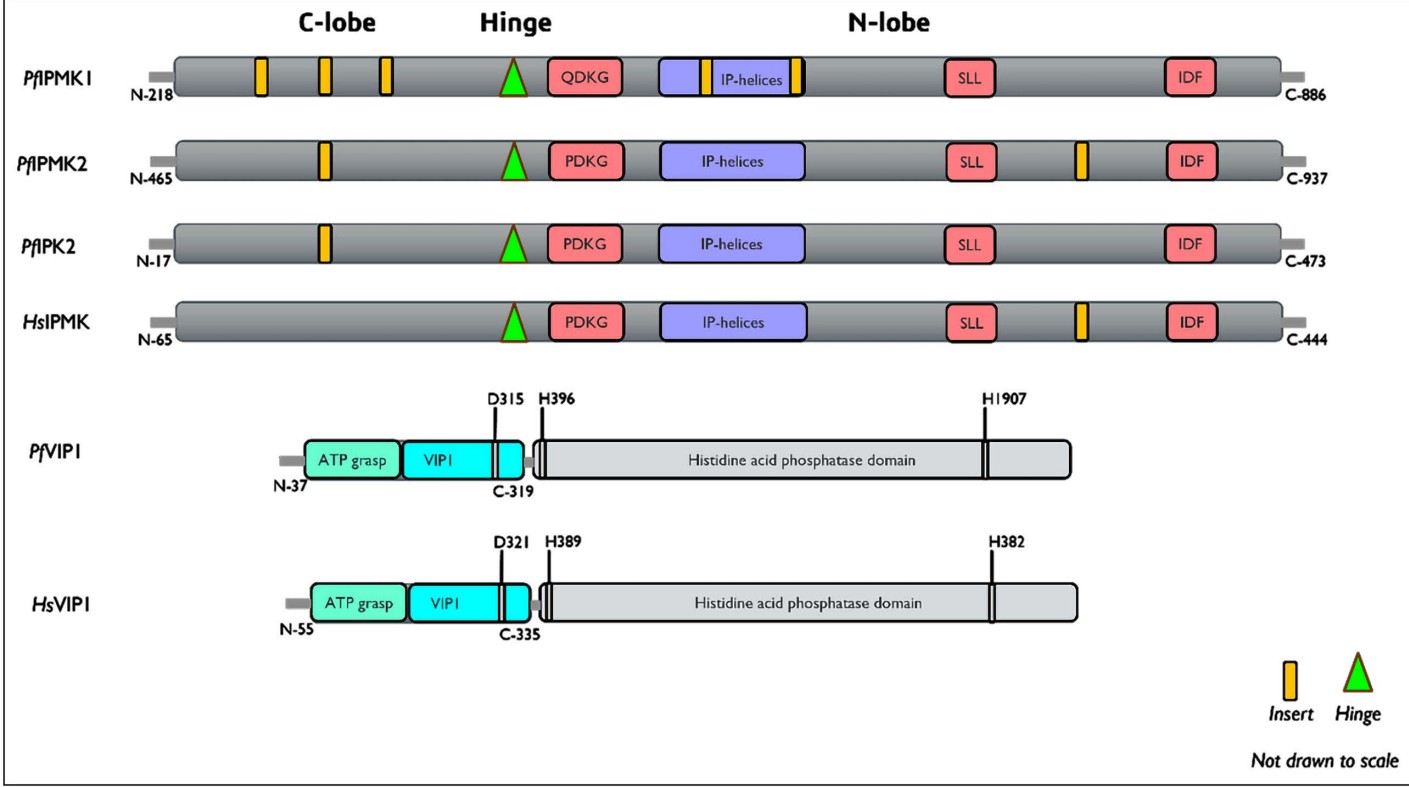

**Fig 4. The domain architecture of putative *P. falciparum* inositol phosphate kinases.** Image showing the functional IPK motifs/residues present in the protein sequences of *P. falciparum* IPKs. In *P. falciparum* VIP1 (*Pf*VIP1), the residue responsible for the catalytic activity is D315, whereas those responsible for the phosphatase activity are H397 and H907. The corresponding human residues are shown in the Fig for *Hs*VIP1.

The differences in similarity percentages between BLASTp and Jalview reflect their distinct alignment strategies: BLASTp reports local similarity in the best-matching region, whereas Jalview calculates global identity across the full kinase domain. For *P. falciparum* VIP1, both methods gave nearly identical similarity values to *H. sapiens* PPIP5K1 (~45%), while the larger discrepancies for *P. falciparum* IPMK2 and IPK2 reflect BLASTp's retrieval of *H. sapiens* IP6K3 as the best local match versus Jalview global comparisons with *H. sapiens* IPMK and IP6K1. Since two similar sequences do not necessarily mean that they are homologous, evolutionary relationships were examined as shown in Figs 5 and 6.

From the phylogenetic tree (Fig 5) it can be inferred that all *P. falciparum* PDKGs are analogous to the human enzymes studied here. Interestingly, *P. falciparum* IP6K shares a common ancestor with yeast IPMKs (Sc/SoIPMKs). When *P. falciparum* IP6K is compared with *H. sapiens* IPK enzymes, both the phylogenetic analysis and percentage sequence identity values indicate that it is more closely related to *H. sapiens* IPMK than to *H. sapiens* IP6Ks — an atypical relationship given its predicted enzymatic classification. Although *P. falciparum* VIP KD, shares 45% identity with *H. sapiens* VIP1, the phylogenetic analysis shows that it is not clustered closely with human VIP1, indicting that it is evolutionarily distinct and not as closely related as sequence alignment alone would suggest (Fig 6).

It seems from the tree generated (Fig 5) that a gene duplication event occurred, resulting in the presence of two IPMKs in *P. falciparum*. *P. falciparum* IPMK1 and IPMK2 are likely paralogous proteins. *P. falciparum* IPMK1 occupies a more basal position within the *Plasmodium* IPMK clade, indicating greater phylogenetic divergence from other eukaryotic IPMKs and carries a QxxxDxKxG catalytic motif substitution, which may impair its kinase function. In contrast, *P. falciparum*

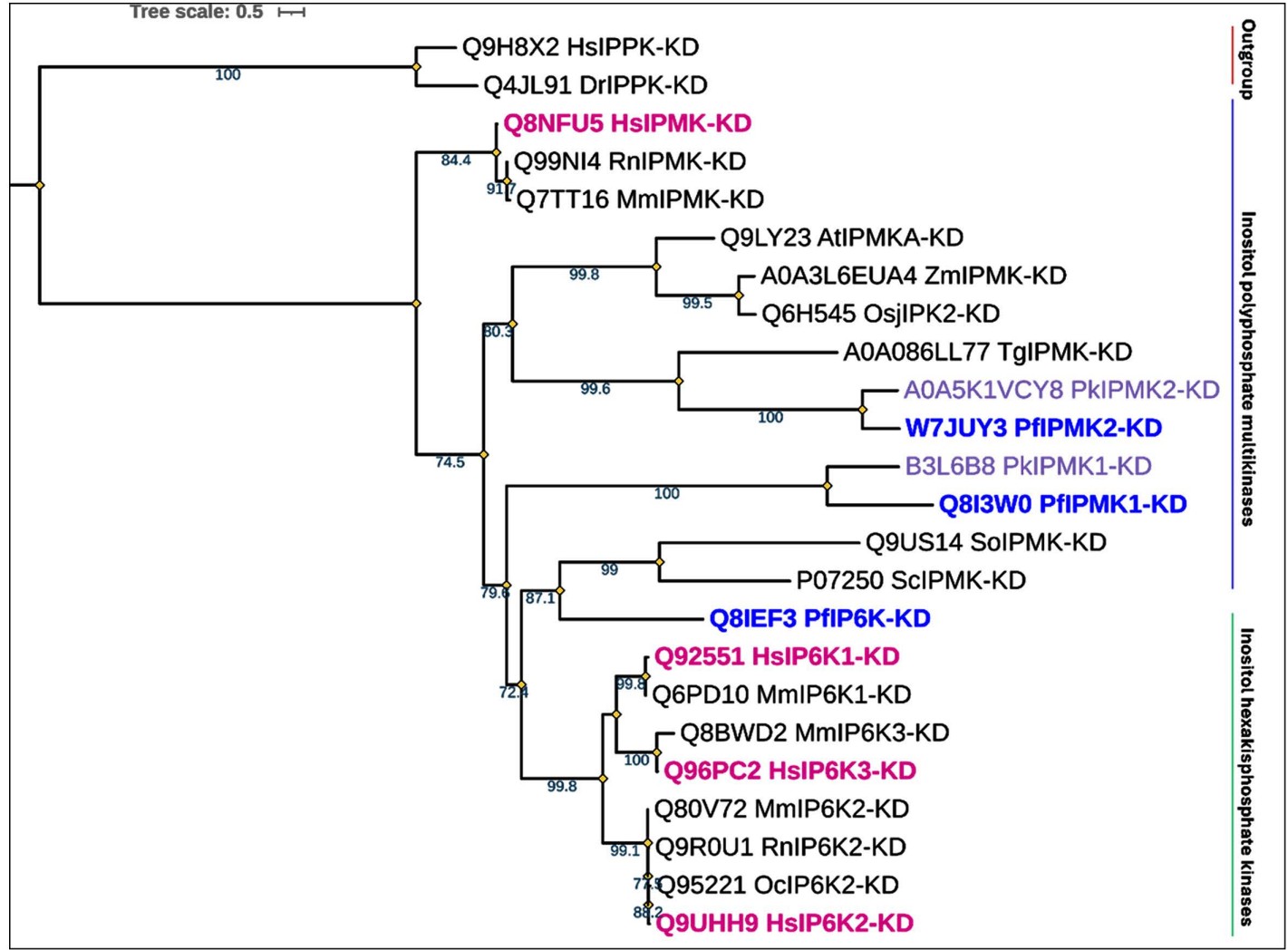

**Fig 5. A rooted phylogeny of kinase domains of inositol phosphate kinases.** Phylogenetic analysis was performed using the IQ-TREE maximum likelihood method. The best fit model LG+F+I+G4 was used for 24 kinase domain sequences with 1326 columns, 966 distinct patterns, 403 parsimony-informative sequences, 457 singleton sites and 465 constant sites. Ultrafast bootstrap support values >70 is shown at the nodes. Tips representing various subfamilies are labelled with inositol pentakisphosphate kinases (IPPKs) as outgroups because IPPKs do not have a conserved PDKG catalytic signature. The *P. falciparum* IPKs (blue labels) are analogous to the *H. sapiens* IPKs (pink labels) but are more closely related to Pk (*P. knowlesi*) enzymes (violet labels). In addition, *Plasmodium* IPMK2s are closely related to *Toxoplasma gondii* (*T. gondii*) IPMK2, consistent with their shared evolutionary lineage as apicomplexans. KD here denotes kinase domains.

IPMK2 retains the canonical PxxxDxKxG motif, suggesting that it may have preserved the ancestral enzymatic activity. *P. falciparum* IPMK2, through domain shuffling or gene fusion, has become part of a larger multidomain protein. This domain may function differently or simultaneously with other domains, such as histone deacetylase II (HDAC2), as previously suggested [74]. This is not surprising, as previous research in other organisms has established a clear link between the activities of these domains.

IPMK and its products IP4 and IP5 have been shown to be involved in the remodelling of chromatin and the regulation of gene expression [75]. Specifically, the abovementioned inositol polyphosphates are able to stimulate the activities of the inositol SWI/SNF (switching-defective/sucrose-nonfermenting) complex, regulate the activities of HDACIII class enzymes,

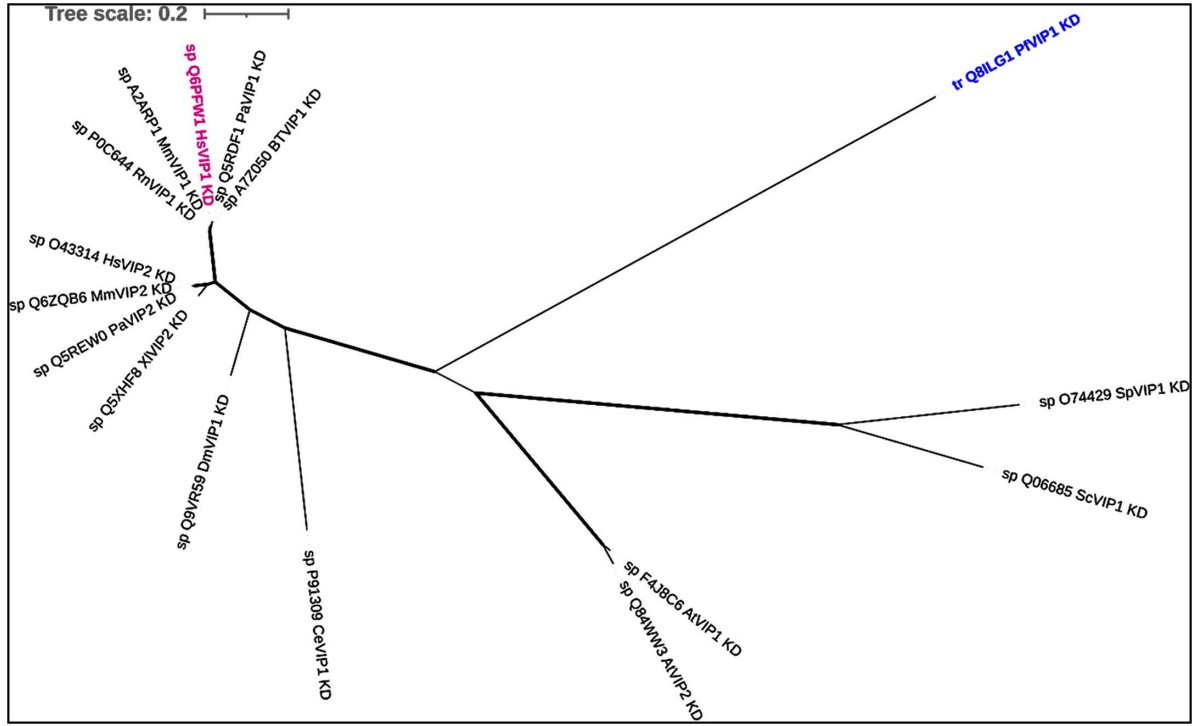

**Fig 6. Unrooted phylogenetic tree of VIP kinase domains (KD).** Analysis of the tree is based on IQ-TREE maximum likelihood. The tree was constructed on the basis of its best-fit model: LG + G4 for 16 protein sequences with 398 columns, 312 distinct patterns, 192 parsimony informative, 90 singleton sites and 116 constant sites. Bold branches have bootstrap support values > 70%. According to this phylogenetic tree, *P. falciparum* VIP_KD is the most divergent among all the sequences being studied, appearing as an outgroup that did not cluster with either of the *H. sapiens* VIP KDs.

such as Sir2A, and are also required for the proper acetylation of histone 4 [75–78]. Thus, an IPK domain close to HDACs may provide a functional advantage for the parasite, consistent with the known hypothesis that multidomain proteins often function cooperatively [79,80].

 *P. knowlesi* IPMK2 has been characterized biochemically and found to have enzymatic activity with a significantly different inhibition profile than *H. sapiens* IPMK [68], suggesting that *P. falciparum* IPMK2 may exhibit similar enzymatic properties. The genes encoding both enzymes are syntenic in *P. falciparum* and *P. vivax* species (data not shown), suggesting that they may be orthologues with a sequence identity of approximately 46%.

**Protein–protein interaction predictions for *P. falciparum* IPKs**

The STRING database is notable for revealing known and predicted protein-protein interactions, which include both physical and functional interactions [45]. After identifying four putative genes encoding enzymes in the IPP of *P. falciparum*, we observed a missing link between the conversion of IP5 to IP7. The mechanism by which *P. falciparum* obtains IP6 to produce pyrophosphates remains unclear, as it lacks the gene encoding inositol-pentakisphosphate 2-kinase (IPPK), the enzyme responsible for converting IP5 to IP6.

 The IPPK is absent in some other eukaryotes too, according to Laha, Portela-Torres [19], who postulated that these organisms either obtain IP6 from their hosts or have an alternate pathway for synthesis. To investigate this, we analysed the protein sequences of all IPKs in *P. falciparum* using the STRING database to identify potential interactions that could provide new biological insights into the pathway in *P. falciparum*.

There was no gene found by STRING network analysis (Fig 7) that might encode an enzyme that produces IP6 in *P. Falciparum*. No predicted interactions were found for *P. falciparum* IPMK1. However, for *P. falciparum* IPMK2, most of the predicted interactions were associated with histone code erasers and their modifying proteins, as discussed earlier. Interpreting the predicted interactions for *P. falciparum* IPMK2 is challenging due to lack of experimental data on the activity of the kinase domain. While the full-length HDACII protein has been extensively studied in *P. falciparum* [74], the specific function of its IPMK domain remains unexplored.

STRING analysis linked IP6K and PPIP5K to upstream/downstream IPP proteins (Fig 7). In *P. falciparum* IP6K, an interesting interaction involved an uncharacterized protein (PF3D7_1309000). Subsequent investigation using the InterPro database [81] revealed that this protein is a member of the nuclear pore complex [GLE1-like superfamily domain], which uses IP6 as a coactivator [82] to facilitate the export of poly(A)+ RNA to the cytoplasm [83,84]. Although these predictions have low confidence levels, they provide valuable insights into the potential functions of these proteins.

## Structural analysis and functional assessment of *P. falciparum* IPKs

In convergent evolution, proteins with distinct evolutionary paths, guided by restricted energetically favourable ways to assemble their secondary structures, arrive at common three-dimensional structures without significant sequence similarity [85,86]. To study this phenomenon, structural alignment was performed. The alignment also helped in identifying active sites and revealing variable regions that can be exploited in the design of selective inhibitors [86–88]. The list of IPK proteins used for structural analysis and their annotations is provided in S1 Table in S1 File.

Structural alignment results (Fig 8) indicate that *P. falciparum* IPMK1 and 2 are structurally distinct from *H. sapiens* IPMK, with RMSD values of 24.0 and 9.2 Å, respectively, but not *P. falciparum* IP6K, which has an RMSD value of 1.0 Å. Surprisingly, *P. falciparum* IP6K is structurally different from *H. sapiens* IP6K, with an RMSD value of 8.2 Å. Among the comparisons, *P. falciparum* VIP1 has the greatest structural resemblance to *H. sapiens* VIP, with the lowest RMSD value of 0.50 Å.

Flexible structural alignment was performed to delve more deeply into the structural similarity between these kinase domains, as it considers the flexibility of the proteins during alignment [48]. The jFATCAT flexible alignment (Table 2) for all residues indicated by sequence length revealed that all *P. falciparum* PDKG domains have distinctive structures but not *P. falciparum* VIP1_N domains. Compared with their human counterparts, *P. falciparum* IPMK1 and IP6K had very low TM scores of 0.05 and 0.13, respectively, which are typical of unrelated proteins. For *P. falciparum* IPMK2, however, the Cα–RMSD between aligned residues was 3.52 Å, and with a TM score of 0.31, it assumes a comparable but nonidentical fold to the *H. sapiens* IPMK, suggesting partial conservation of tertiary structure and potential retention of ancestral activity.

Both *P. falciparum* IPMKs show promising characteristics as drug targets, with the exception that *P. falciparum* IPMK1 has a mutation in which the Pro is substituted for Gln in its catalytic motif, as previously shown. To scrutinize both *P. falciparum* IPMKs as drug targets, *in silico* functional assessment was performed through molecular docking. Before docking, the predicted structures of *P. falciparum* IPMK1 and IPMK2 were validated for quality and accuracy (S2 and S3 Figs in S1 File). AlphaFold's highest scoring model for *P. falciparum* IPMK1 identified five core ATP-binding residues that take part in hydrogen bonding: Lys$^9$, Glu$^{364}$, Val$^{366}$, Asp$^{379}$ and Asp$^{624}$, as well as other residues: Lys$^{365}$, Gln$^{375}$, Arg$^{556}$ and Thr$^{626}$. For *P. falciparum* IPMK2, the ATP-binding residues predicted were Lys$^{14}$, Ser$^{204}$, Val$^{206}$, Asp$^{217}$, and Asp$^{442}$, with an additional residue, Trp$^1$ (Figs 9 and 10).

## The structural basis of hinge architecture and ATP binding in *P. falciparum* IPMKs

To gain deeper understanding of how hinge organization affects the binding of ATP and the druggability of *P. falciparum* IPMKs, we examined the structural basis of hinge architecture and ATP coordination.

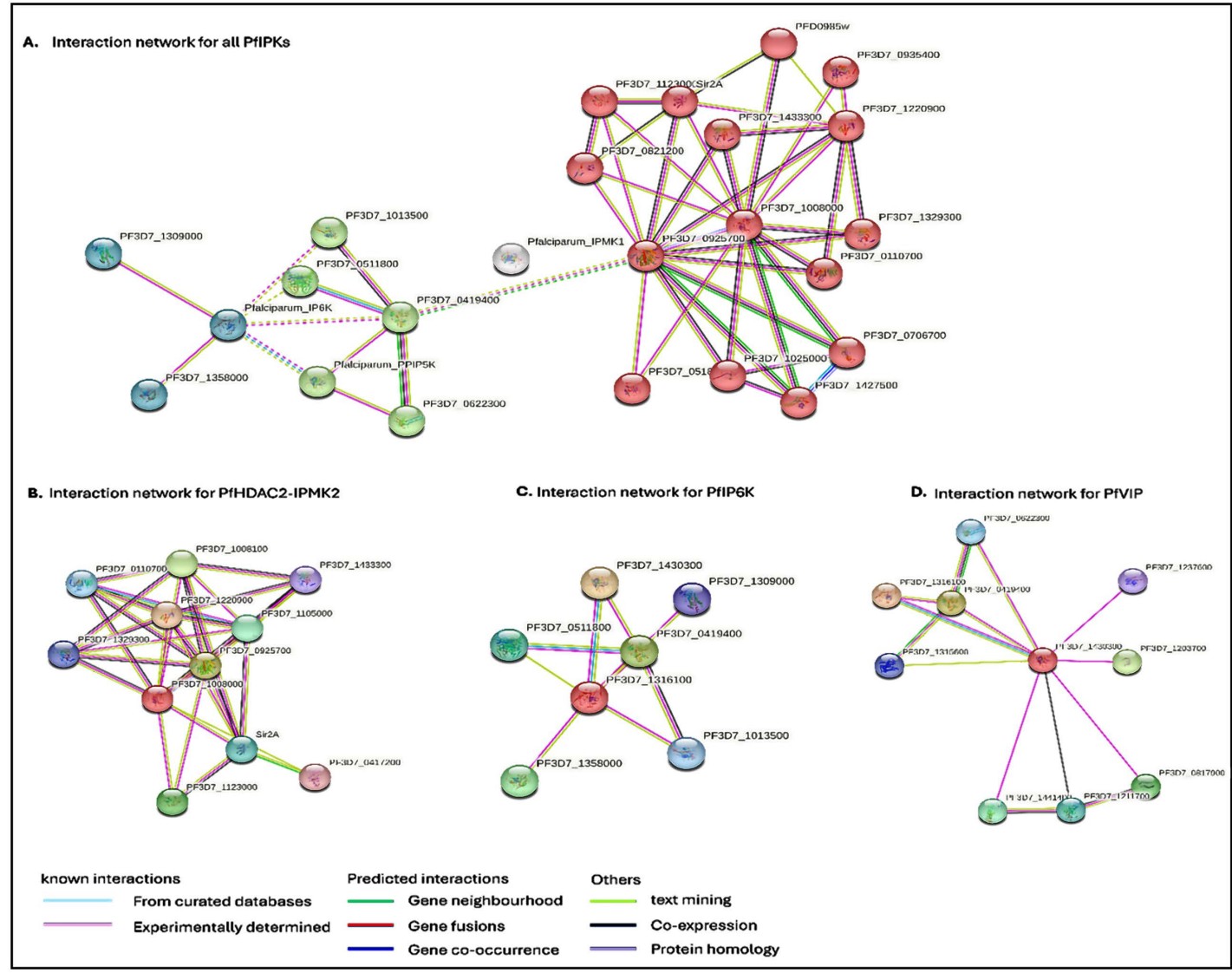

**Fig 7. STRING interaction network analysis of *P. falciparum* IPKs.** (A) All putative protein sequences of *P. falciparum* IPKs were subjected to a STRING database search for pathway analysis and clustered with MCL clustering, in which each cluster is coloured to denote potential interactions of *P. falciparum* IPMK1 (white), IPMK2 (red), IP6K (teal) and PPIP5K (green). No interaction was predicted for *P. falciparum* IPMK1, although a medium confidence score of ≥ 0.4 is shown. (B) Protein-protein interactions of the red node PF3D7_1008000 (HDAC2_IPMK2) from the predicted network are closely associated with histone deacetylase 1 (PF3D7_0110700), repressive histone modifying enzymes such as itself, including chromatin assembly factors and histone binding proteins (PF3D7_1433300, Sir2A, PF3D7_1329300), and histone 4 (PF3D7_1105000). Interactions also included heter-ochromatin protein 1 (PF3D7_1220900), which is a repressive histone-associated protein, and then a *Plasmodium*-specific uncharacterized protein (PF3D7_1123000). (C) *P. falciparum* IP6K (red node) possibly interacts with phospholipase C (PF3D7_1220900), putative patatin-like phospholipase (PF3D7_1220900), inositol 3-phosphate synthase (PF3D7_0511800), *P. falciparum* PPIP5K (PF3D7_1430300) and two uncharacterized proteins (PF3D7_1309000 and PF3D7_0419400). (D) *P. falciparum* PPIPK (red node) was predicted to interact with *P. falciparum* IP6K, uncharacterized protein (PF3D7_0419400), vacuolar transporter chaperone, putative (PF3D7_0622300), high mobility group protein B2 (PF3D7_0817900), nucleo-some assembly protein (PF3D7_1203700), DNA replication licensing factor MCM5 putative (PF3D7_1211700), periodic tryptophan protein 1, putative (PF3D7_1237600), CDP-diacylglycerol-inositol 3-phosphatidyltransferase (PF3D7_1315600) and a component of the FACT complex; FACT complex subunit SSRP1 (PF3D7_1441400).

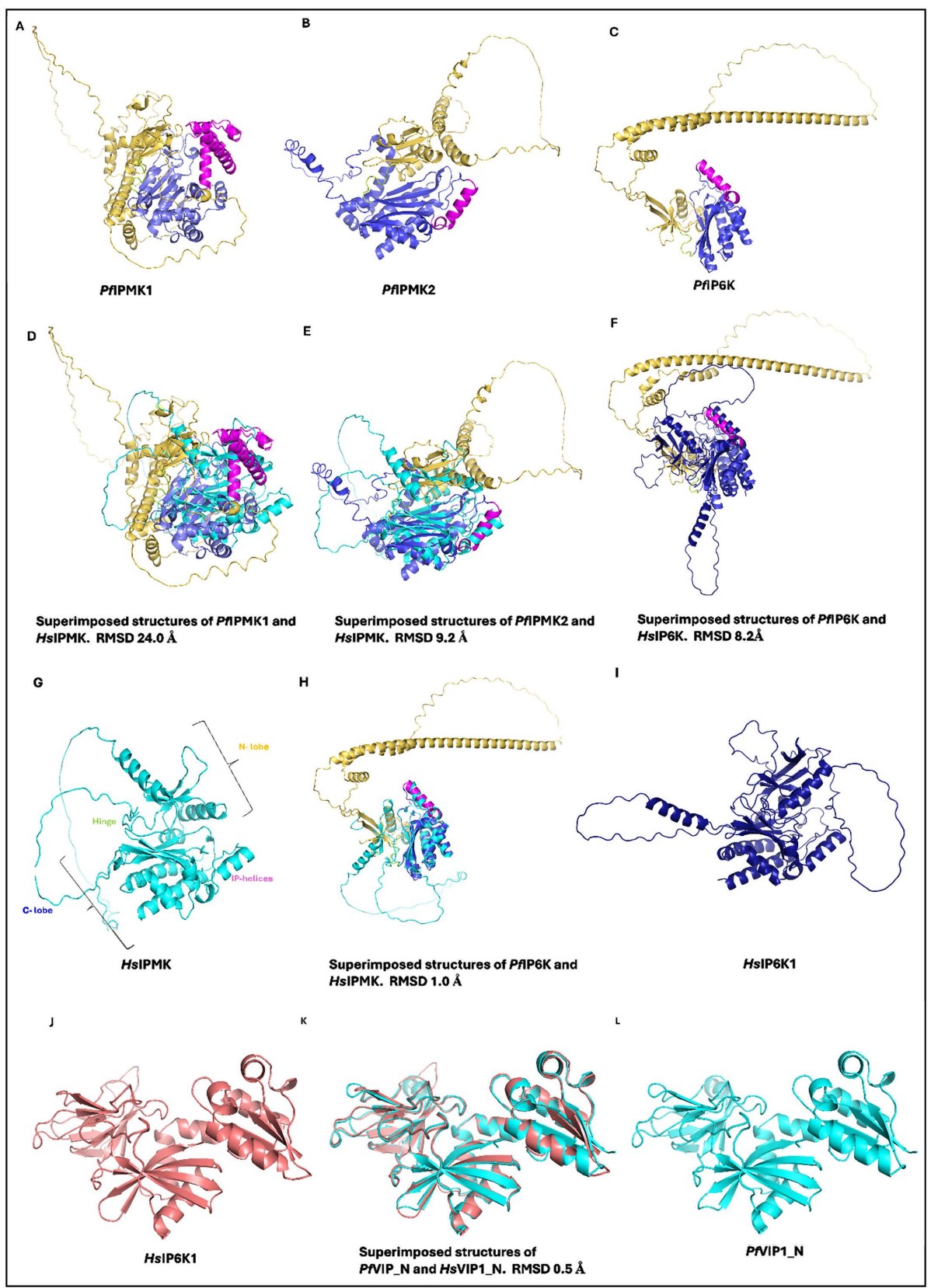

**A** *Pf*IPMK1

**B** *Pf*IPMK2

**C** *Pf*IP6K

**D** Superimposed structures of *Pf*IPMK1 and *Hs*IPMK. RMSD 24.0 Å

**E** Superimposed structures of *Pf*IPMK2 and *Hs*IPMK. RMSD 9.2 Å

**F** Superimposed structures of *Pf*IP6K and *Hs*IP6K. RMSD 8.2Å

**G** N- lobe / Hinge / IP-helices / C- lobe / *Hs*IPMK

**H** Superimposed structures of *Pf*IP6K and *Hs*IPMK. RMSD 1.0 Å

**I** *Hs*IP6K1

**J** *Hs*IP6K1

**K** Superimposed structures of *Pf*VIP_N and *Hs*VIP1_N. RMSD 0.5 Å

**L** *Pf*VIP1_N

**Fig 8. Analysis of kinase domains of *P. falciparum* inositol phosphate kinase superfamily members.** A, B, C and J are the structures of the kinase domains of *P. falciparum* IPKs predicted using AlphaFold and the aligned structures of *P. falciparum* and *H. sapiens* IPKs, with their RMSD shown in D, E, F, H and K. G, I and L are *H. sapiens* IPKs. Different parts of the kinase domain are colored differently: the orange cortoons represent the N-lobe, the purple-blue cortoons represent the C-lobe, the magenta cortoon represents the IP-helices, and the limon represents the hinge region. Structural analysis and alignments were performed in PyMOL.

In canonical kinases, ATP binding is mediated by a bilobal catalytic core, where its adenine ring fits snugly into the hinge region that connects the N- and C-lobes [90]. Although IPMKs belong to a distinct family of IPKs, they employ a similar strategy for ATP binding and the substrate binding IP helices are inserted into the C-lobe [22,53,91]. A hallmark of IPMKs is the conserved PxxxDxKxG motif [65,92,93]; here, the terminal proline of the hinge may confer local rigidity favourable for ATP coordination. The rigid five-membered ring of proline residue restricts the $\varphi$ dihedral angle and reduces local conformational entropy [94], thereby probably stabilizing hinge-adjacent geometry.

In human IPMK, Pro[140] appears to help enforce this structural restraint thereby, enabling Glu[131] and Val[133] to anchor the adenine base via conventional hydrogen bonding [53]. In *P. falciparum*, however, IPMK1 harbours a Gln[375] substitution at the position equivalent to the conserved proline residue found in canonical IPMKs. This substitution likely removes the conformational constraint normally imposed by proline, increasing local flexibility and potentially misaligning the hinge backbone. Structural alignment with human IPMK shows a pronounced global RMSD of ~24 Å (Fig 8D) and a hinge-specific RMSD of 0.806 (Fig 11A) indicating distortion of the ATP-binding interface. These deviations are confined to the hinge region, while the flanking β-strands that form the base of the active site remain well aligned (Fig 9B), linking motif alteration specifically to hinge misalignment and potential disruption of adenine base recognition during ATP binding (Fig 11D). Thus, several of the conserved positions we align do fall within the ATP-binding architecture (hinge and catalytic residues). However, our selectivity rationale does not rely on these conserved adenine-contacting atoms; rather, it derives from the non-conserved pocket microenvironment (parasite-specific wall residues and shape) and adjacent allosteric cavities, which differ between *P. falciparum* IPMK2 and human IPMK.

A 2D interaction analysis in Discovery Studio shows that Gln[375] does not participate in adenine binding, and the ATP base forms only a single hydrogen bond with Val[366] at the hinge (Figs 9B, 11D). This limited hinge interaction contrasts with canonical kinases or IPMKs, where two hinge residues typically anchor ATP via hydrogen bonds. Although PyMOL suggested a Gln[375]-ATP hydrogen bond, this was absent in Discovery Studio, likely due to marginal geometry. ATP binding in *P. falciparum* IPMK1 instead relies mainly on electrostatic and non-hinge hydrogen bonding; docking predicts a comparable binding affinity of −6.0 kcal/mol (Fig 9), though canonical hinge geometry is not restored.

**Table 2. Results of structural superimposition of *H. sapiens* and *P. falciparum* IPKs. The TM score is in the range of 0–1, where 1 indicates a perfect match between two structures [89].**

| Entry | RMSD (Å) | TM-Score | Identity (%) |
|---|---|---|---|
| *H. sapiens* IPMK_KD | – | – | – |
| *P. falciparum* IPMK1_KD | 2.88 | 0.05 | 13 |
| *P. falciparum* IPMK2_KD | 3.52 | 0.31 | 18 |
| *P. falciparum* IP6K_KD | 2.41 | 0.09 | 21 |
| *H. sapiens* IP6K_KD | – | – | – |
| *P. falciparum* IP6K_KD | 2.37 | 0.13 | 28 |
| *H. sapiens* VIP1_KD | – | – | – |
| *P. falciparum* VIP1_KD | 1.37 | 0.96 | 48 |

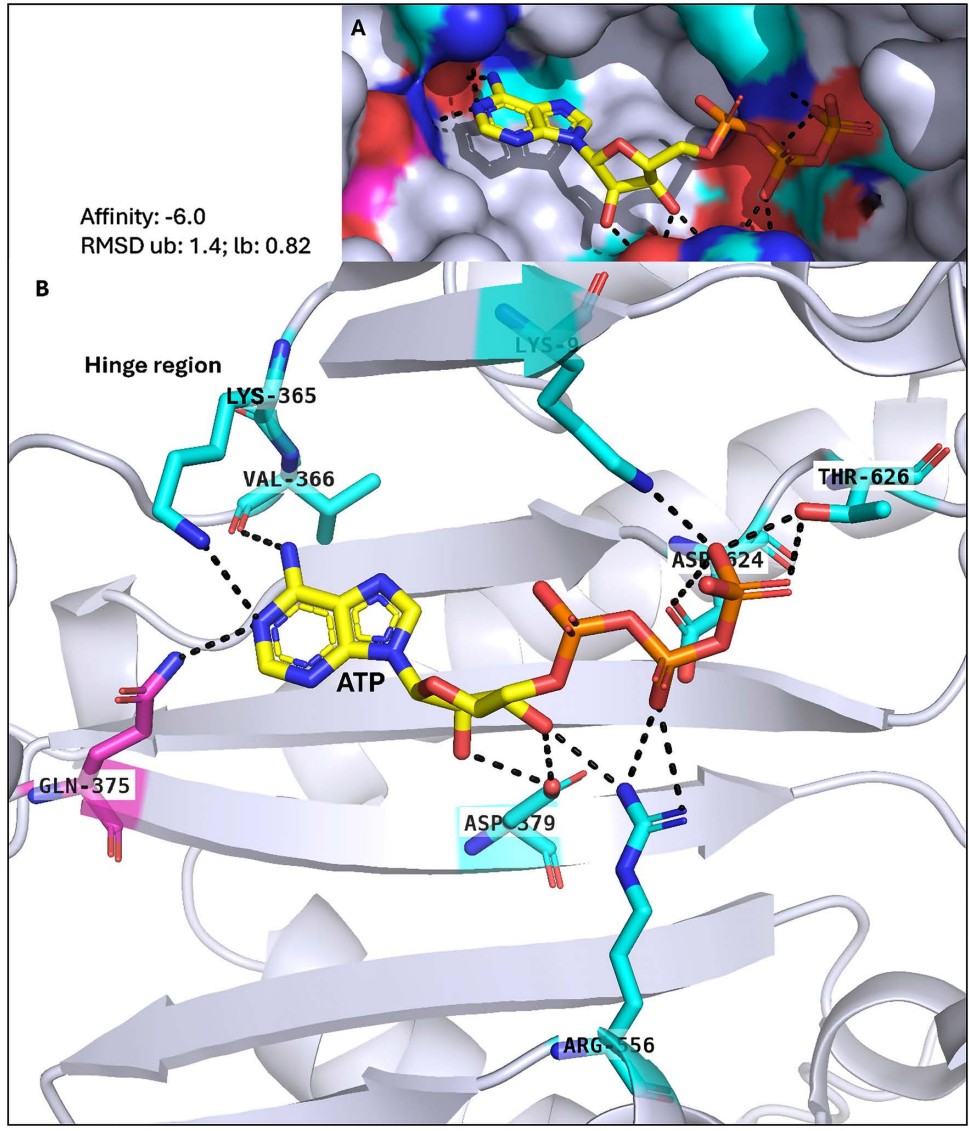

Affinity: -6.0
RMSD ub: 1.4; lb: 0.82

**Fig 9. Functional assessment of *P. falciparum* IPMK1.** *P. falciparum* IPMK1-ATP interaction: (A) is the surface view and the cartoon view (B). ATP was docked into the kinase domain of *P. falciparum* IPMK1 using PyRX, and its interactions with the active residues were analysed in PyMOL. The cyan sticks represent active residues, and the magenta sticks represent mutations in the PDKG motif (Pro398Gln) affecting ATP binding.

In contrast, ATP binding in *P. falciparum* IPMK2 is driven primarily by hinge residues SER[204] and VAL[206], which form direct hydrogen bonds with the ATP base. A 2D interaction plot of human IPMK (Fig 11E; PDB: 5W2H) shows an almost identical interaction pattern, with Glu[131] and Val[133] anchoring the ATP adenine base via hydrogen bonds. These conserved interactions validate the accuracy of the *P. falciparum* IPMK2 docking predictions (Fig 10), which yielded a docking score of −6.1 kcal/mol and support a hinge-centred ATP recognition mechanism. Structural superimposition reveals a lower overall RMSD of ~9 Å (Fig 8E) with a hinge RMSD of 0.424 (Fig 11E), which supports a stable hinge structure conducive to the recognition of adenine base during ATP binding. Its 2D interaction plot generated in Discovery Studio also confirmed Pymol-based docking analysis, showing a complete ATP-binding profile including hydrogen bonds with Asp[217], Trp[1],

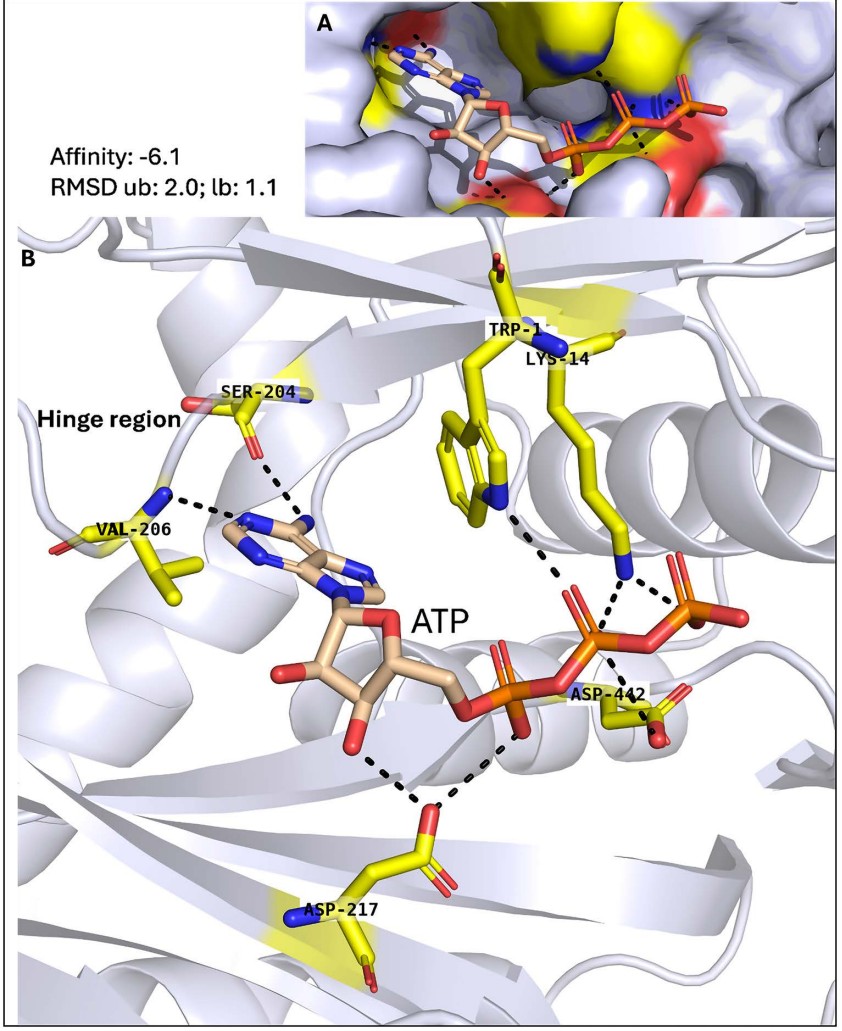

**Fig 10. Functional assessment of *P. falciparum* IPMK2.** (A) A surface representation showing ATP binding and (B) a cartoon representation of the predicted *P. falciparum* IPMK2 domain with ATP docked into its active pocket, showing residues (yellow) in the active pocket that bind to ATP. The black dotted lines represent hydrogen bonds between interacting atoms.

and Lys[14] (also forming a salt bridge and attractive charge interaction). These non-covalent interactions further stabilize the pocket.

Beyond ATP coordination, *P. falciparum* IPMK2 substrate-binding residues were also predicted to be Lys[219], Ser[233], Lys[236], Lys[237], Lys[240], Arg[244], His[275] and His[445] and with more flexible amino acids, such as Lys, rotamers of the side chains may provide wider substrate promiscuousness and specificity than the human IPMK substrate domain, which contains a series of Gln residues [53].

These *in silico* structural data show that the Gln[375] substitution in *P. falciparum* IPMK1 disturbs the hinge alignment and affects the binding of adenine base, while *P. falciparum* IPMK2 preserves a conserved, catalytically viable ATP-binding site. This difference highlights the structural viability of *P. falciparum* IPMK2 as a therapeutic target, in contrast to the distorted and noncanonical binding site of *P. falciparum* IPMK1. Since IPMK2 is viable, we herein report it as *P. falciparum* IPMK.

**A.** *Hs*IPMK (magenta) vs. *Pf*PMK1 (yellow) – distorted hinge
(RMSD 0.806 Å)

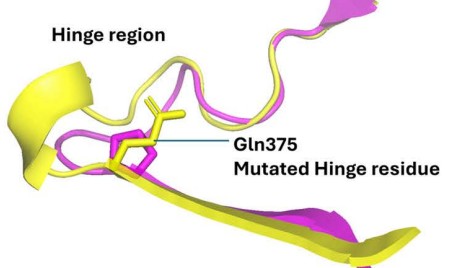

**B.** *Hs*IPMK (magenta) vs. *Pf*PMK2 (blue) – conserved hinge
(RMSD 0.424 Å)

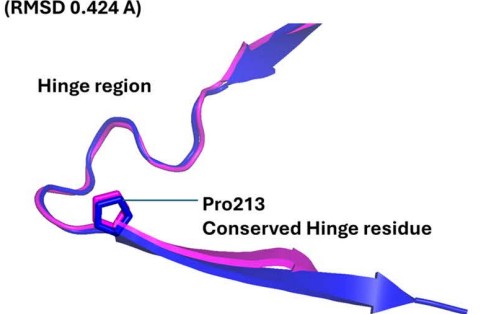

2D ligand–protein interaction diagrams of ATP bound to IPMKs.

**C.** *Hs*IPMK-ATP

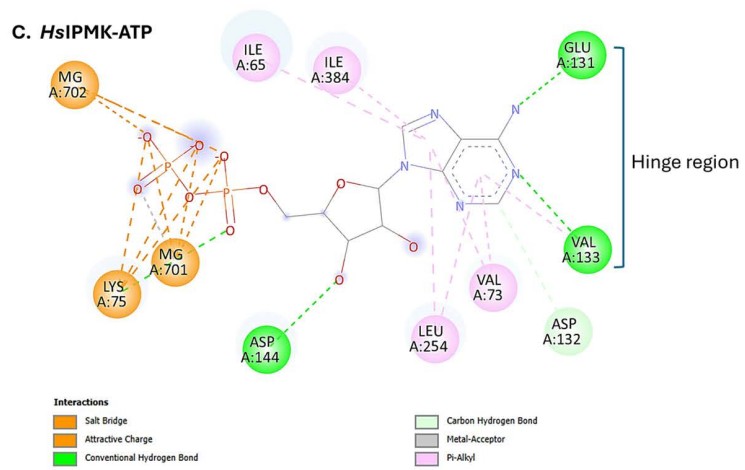

**D.** *Pf*PMK1-ATP          **E.** *Pf*PMK2-ATP

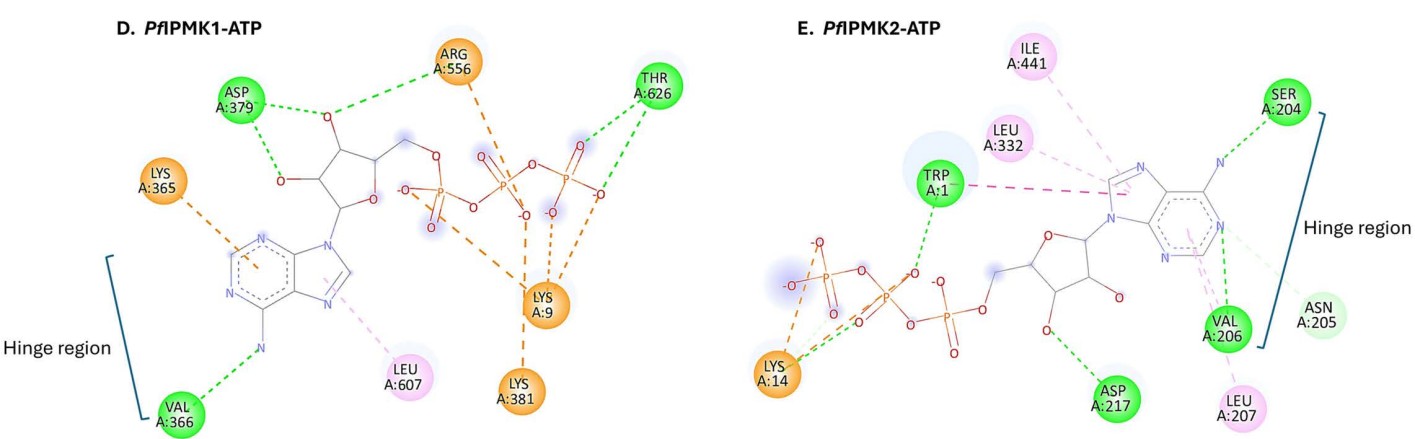

**Fig 11. Hinge-specific alignment and ATP-binding interactions of *P. falciparum* and *H. sapiens* IPMKs.** Structural superimposition of the hinge region of *H. sapiens* IPMK (magenta) with (A) *P. falciparum* IPMK1 (yellow), showing distortion due to the Gln[375] (stick representation) substitution and (B) *P. falciparum* IPMK2 (blue), showing well-aligned hinge geometry and conserved proline residues (stick representation). (C) 2D interaction plot of *H. sapiens* IPMK (PDB: 5W2H) with ATP, showing hinge hydrogen bonds between Glu[131], Val[133], and the adenine base. (D) 2D interaction plot of *P. falciparum* IPMK1 with ATP, showing a single hinge hydrogen bond between Val[366] and the adenine base; Gln[375] does not interact. (E) 2D interaction plot of *P. falciparum* IPMK2 with ATP, showing conserved hinge hydrogen bonds between Ser[204], Val[206], and the adenine base, along with additional interactions with Asp[217], Trp[1], and Lys[14]. The interaction profile is similar to that of the human IPMK–ATP complex. The green dotted lines represent hydrogen bonds between interacting atoms.

## Druggability prediction of *P. falciparum* IPMK

In computer-aided drug discovery, a critical step after target prediction is identifying and characterizing druggable pockets. PockDrug predicted 5 druggable pockets in *P. falciparum* IPMK with druggability scores ranging from 0.58 to 1.00 (S2 Table in S1 File). These included the ATP-binding site (P3) and an allosteric pocket near the hinge region (P8) as well as other promising sites that could be explored for selective inhibition (Fig 12).

Structural differences between *P. falciparum* and *H. sapiens* IPMK support selective drug targeting. Unlike *H. sapiens* IPMK, *P. falciparum* IPMK lacks a proline-loop crucial for substrate binding [92], and has a C-lobe insertion in addition to extra alpha helices in its N-lobe (Fig 12) making it structurally unique. These unique features underscore the potential for developing selective inhibitors that specifically target *P. falciparum* IPMK while minimizing off-target effects on human kinases.

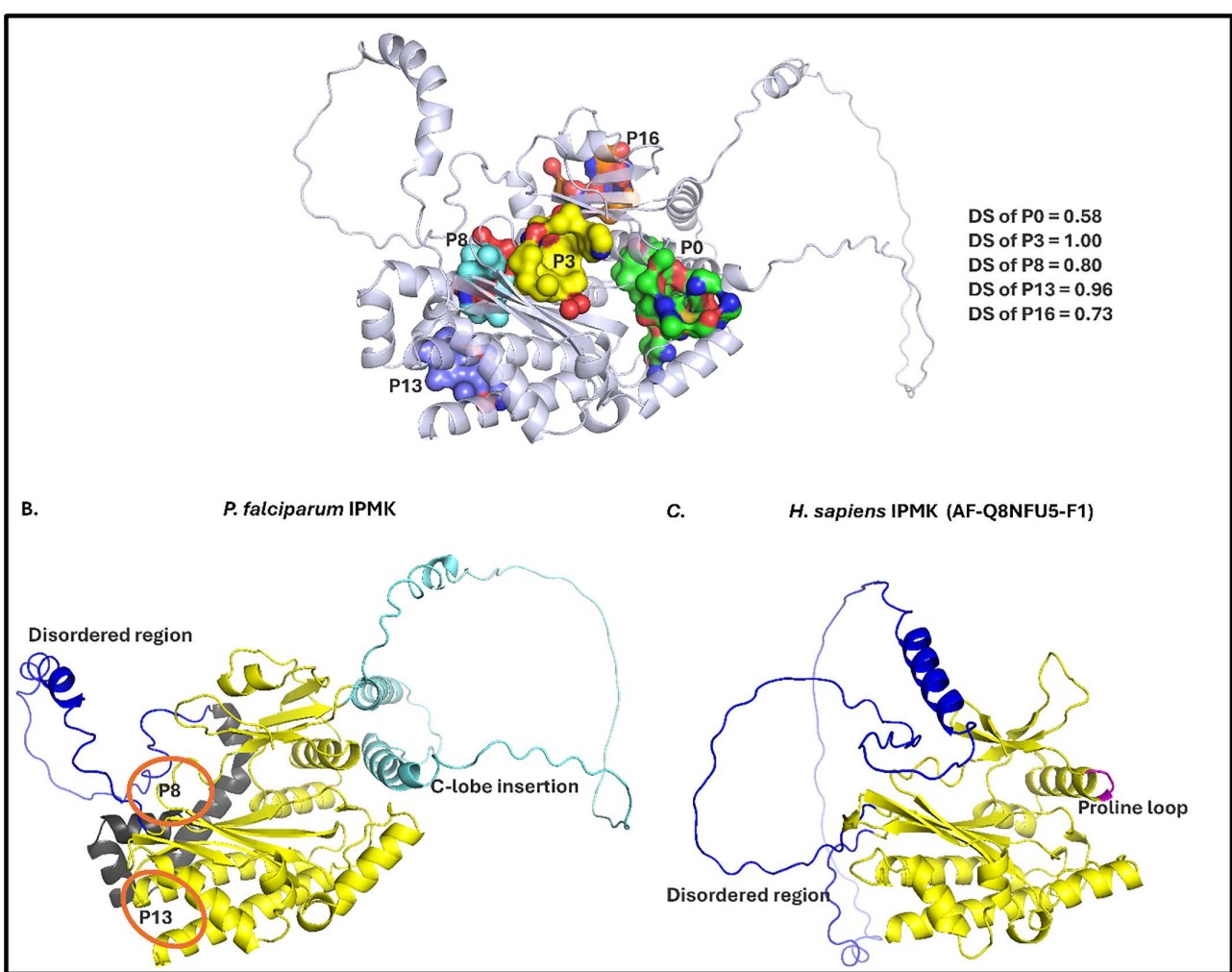

**Fig 12. PockDrug druggable pocket predictions for *P. falciparum* IPMK2.** A. Visualization of the 5 druggable pockets (P) predicted for *P. falciparum* IPMK2 with their corresponding druggability scores (DSs). The scores range from highly druggable (DS 1.0) to non-druggable (DS 0), and DS > 0.5 indicates a druggable pocket. B is the distinctive 3D structure of *P. falciparum* IPMK with its unique pockets (orange circles) created because of extra helices in its N-lobe (black coloured helices). It has a relatively shorter disordered region and extra insertion in its C-lobe (cyan coloured). C. the human structure (AF-Q8NFU5-F1) has a longer disordered region, and a unique proline loop (magenta coloured) which is also absent in *P. falciparum* IPMK.

## Predicted subcellular localization, functions of *P. falciparum* IPKs and its inositol (pyro)phosphate pathway

Subcellular localization plays a critical role in inferring enzyme functions and assessing druggability [95,96]. As summarized in S3 Table in S1 File, *P. falciparum* IPMK2 and IP6K were predicted to localize in both the nucleus and cytoplasm, while *P. falciparum* PPIP5K was annotated as a cytoplasmic protein. The functional predictions of these proteins, together with experimentally determined functions of inositol phosphate kinases from the literature, have been integrated into our proposed model of the *Plasmodium falciparum* inositol pyrophosphate pathway (Fig 13).

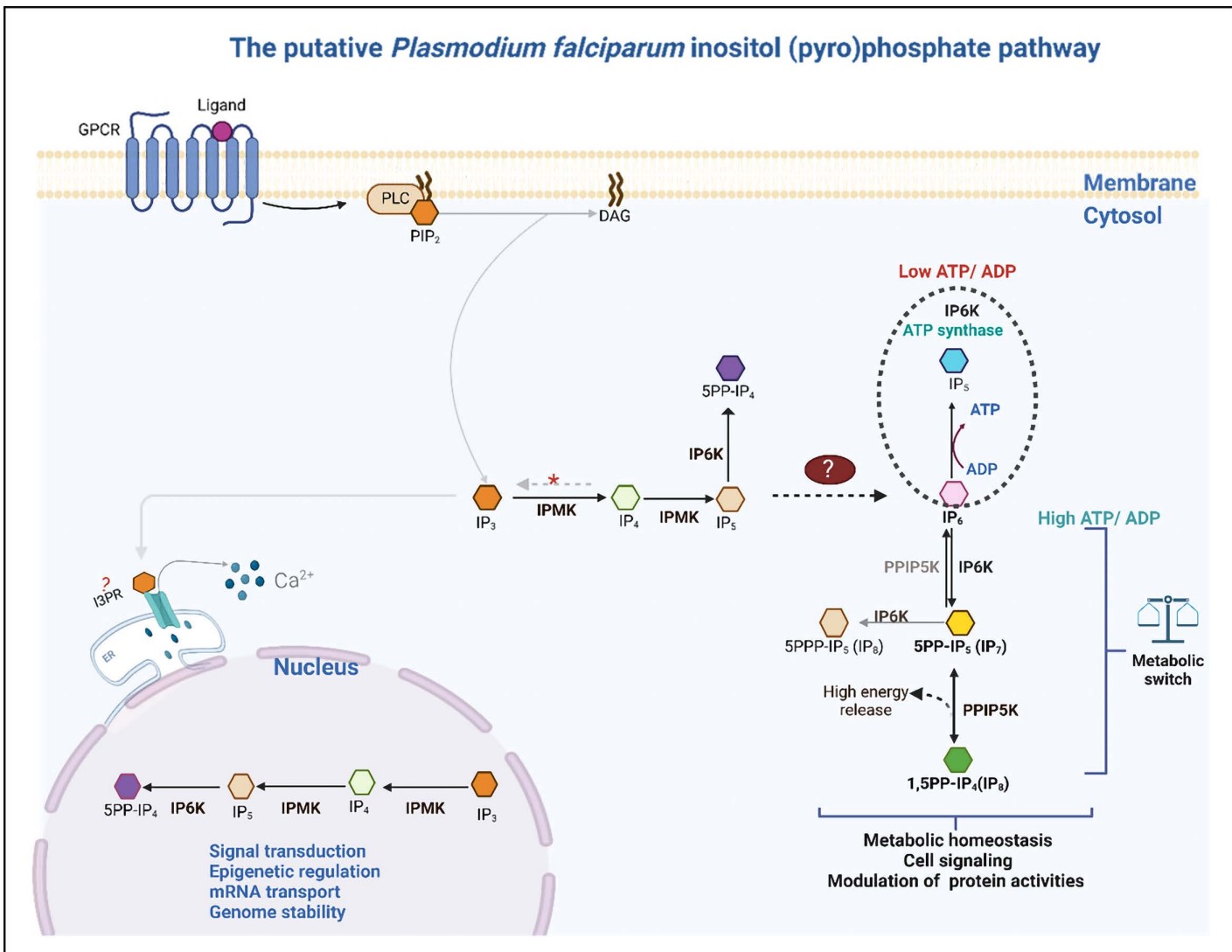

**Fig 13. Putative *P. falciparum* inositol pyrophosphate pathway.** The simplified pathway begins with the hydrolysis of PIP$_2$ to IP$_3$ and DAG. *P. falciparum* IPMK sequentially phosphorylates IP$_3$ to IP$_5$. The putative enzyme that dephosphorylates IP4 to IP3 is indicated by a red asterisk. Pyrophosphate synthesis begins when *P. falciparum* IP6K phosphorylates IP6 to 5PP-IP$_5$ (5-diphosphoinositol pentakisphosphate) (IP$_7$), which is further phosphorylated to 1,5PP-IP$_4$ (1,5-bisdiphosphoinositol tetrakisphosphate) (IP$_8$) [20,25,26,28,30,71,73,97–100]. According to predictions of the subcellular localizations of the *P. falciparum* IPKs, pyrophosphate synthesis only occurs in the cytosol, which is in agreement with what has been reported in the literature [101]. On the basis of events that occur in other organisms studied, it may be possible for *P. falciparum* IP6K with a low ATP/ADP ratio to synthesize ATP from ADP and IP6 to generate energy [102], and at a high ATP/ADP ratio, the production of pyrophosphates is favoured [28,69]. IPMK has also been found to regulate the transport of mRNAs [103]. Fig created with BioRender.com.

## Confirmation of the sequences of *P. falciparum* IPKs

This study presents the first *in silico* characterization and experimental validation confirming the identities of *P. falciparum* inositol (pyro)phosphate kinases: IPMK1, IPMK2, IP6K, and VIP1. While the genes encoding these enzymes are already annotated in PlasmoDB and other reference databases, our aim was to experimentally verify the p.Pro375Gln mutation in IPMK1 and confirm sequence integrity of the other IPKs in both laboratory strains and a field isolate (Fig 14).

Illumina MiSeq amplicon sequencing confirmed the gene sequences matched those deposited in the PlasmoDB database [34]. Because we were particularly interested in validating the p.Pro375Gln mutation detected in *P. falciparum* IPMK1, the DNA sequence obtained from the field isolate was translated into its corresponding protein sequence using the ExPASy translation tool [104]. The ExPASy output (see S1 Sequence in S1 File) confirmed the presence of this mutation.

## Expression of *P. falciparum* IPKs across the Life Cycle

We examined the mass spectrometry-based proteomic data deposited in PlasmoDB [34] to study the expression patterns of these enzymes at various stages of the *P. falciparum* life cycle. Due to methodological differences in data collection, only qualitative data was considered for the analysis. These could shed light on the functional roles of these enzymes and their products at various life cycle stages and possibly contribute to the survival of the parasite. Peptides mapped to inositol phosphate kinases were enriched in samples from the erythrocytic, gametogenesis, oocyst and salivary sporozoite stages (S4 Table in S1 File) [105–112]. These observations suggest that the inositol pyrophosphate pathway is likely relevant at multiple stages of the life cycle of the parasite life cycle.

We note that PF3D7_1008000 encodes a multidomain, intronless protein containing a mid-terminal HDAC2 domain fused to a C-terminal IPMK domain [74]. No alternative splicing events have been reported for this locus, so detection of transcripts from the gene in RNA-seq datasets directly implies expression of both domains. Published proteomic studies have detected peptides from the PF3D7_1008000 gene product across multiple life cycle stages [105–112]; while these detections are not domain-specific, they confirm translation of the fusion protein in diverse developmental contexts. Because the HDAC2 domain is embedded within the intact fusion, localization data from HDAC2-tagged constructs likely reflect the distribution of the full-length HDAC2–IPMK protein, barring proteolytic cleavage [74]. Functional genetic studies targeting the full-length gene demonstrate that this HDAC2–IPMK fusion is indispensable for parasite proliferation, though they do not resolve whether essentiality derives primarily from one domain or from their coupled function [74]. Nevertheless, given the stage-spanning transcript and protein evidence, and the established nuclear roles of IPMKs in chromatin regulation in other systems, it is reasonable to hypothesize that the IPMK domain contributes materially to the essential function of the fusion protein.

## Metabolic and developmental roles of the inositol pyrophosphate pathway

The metabolic activity of the *P. falciparum* malaria parasite is complex and fluctuates to meet the demands of its developmental transitions. The inositol pyrophosphate pathway may contribute to parasite adaptation during these changes. During the less metabolically active ring stage, where the cytosolic ATP/ADP ratio is high, pyrophosphate synthesis may be favoured. As the parasite transitions to the trophozoite and schizont stages, a lower ATP/ADP ratio may promote pyrophosphate dephosphorylation, providing energy for actively proliferating cells [24,113]. Pyrophosphates have been established as energy sources in *P. falciparum* ring and trophozoite stages [114].

## Nuclear functions and epigenetic regulation

Apart from pyrophosphate synthesis, most of the intermediate inositol phosphates produced by these inositol phosphate kinases in the nucleus are involved in gene regulatory functions, DNA repair, mRNA export, protein synthesis and even epigenetic mechanisms (Fig 13) [66,75–77,103,115,116]. In view of this, the IP products generated by the *P. falciparum*

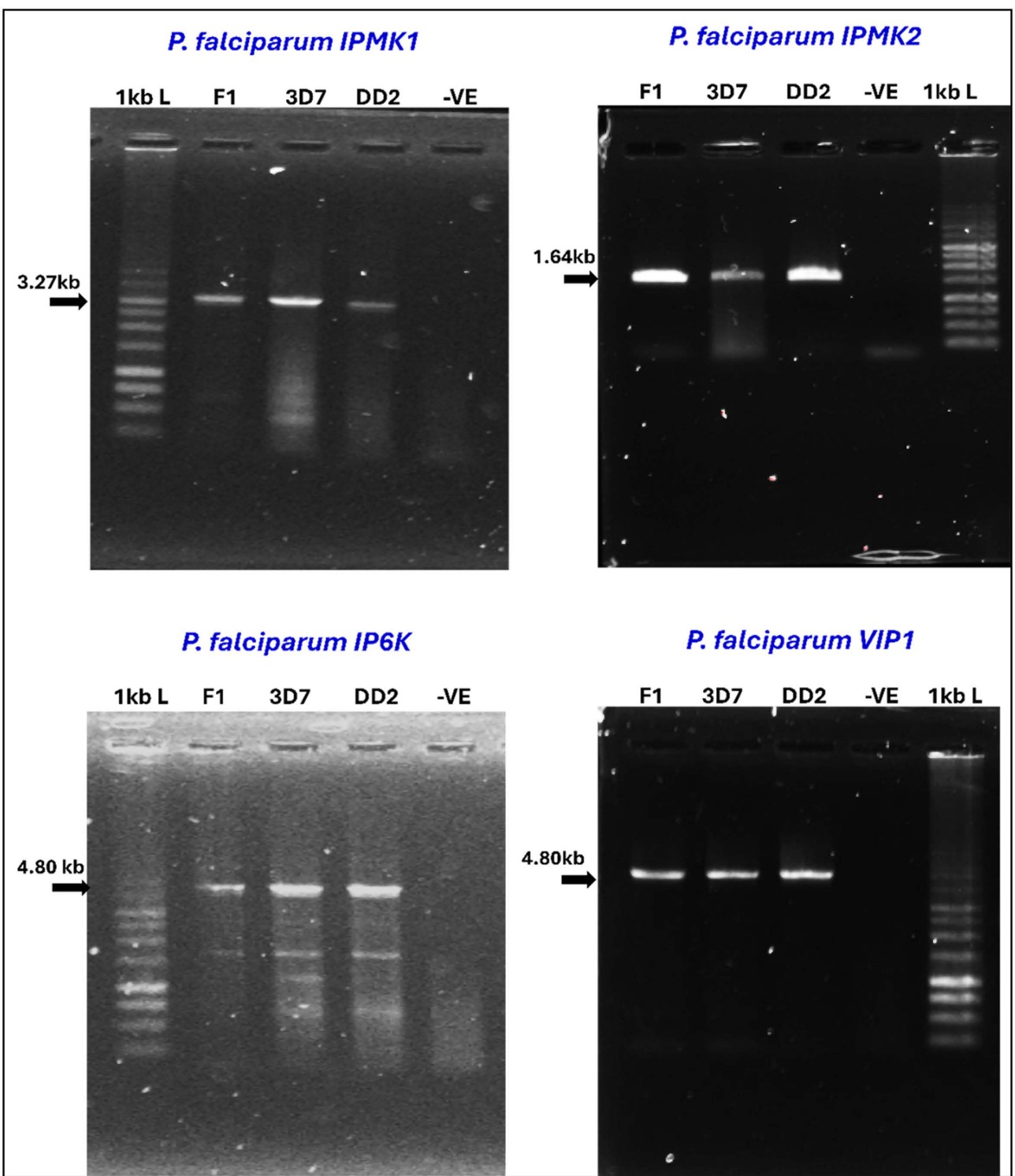

**Fig 14. Conventional PCR amplification of *P. falciparum* IPKs.** Agarose gel electrophoresis analysis for the detection of the 4 genes encoding *P. falciparum* IPKs in one field isolate (F1) and the laboratory strains 3D7 and DD2 (A & B) using specific primers for the genes. Bands were analysed with a 1 kb ladder (L) from Promega; lanes for the negative control for PCR are labelled -VE.

nuclear enzymes IPMK and IP6K may be vital in controlling the transcription of genes and transducing signals needed at specific times and in response to environment changes.

It is highly possible that *P. falciparum* IPMK could contribute to the transcriptional regulation of the var genes, both directly and indirectly, through epigenetic mechanisms [117]. Since var gene expression is controlled by histone modifications and chromatin accessibility, and inositol phosphates are known to modulate chromatin remodellers, we hypothesize that IPMK-generated phosphates influence var gene switching.

### Interaction with HDACs and implications for targeting

The decision to commit sexually or not is also epigenetically regulated, which can be influenced by inositol phosphates and pyrophosphates. In humans, IP4 plays an essential role in stabilizing the deacetylase activation domain (DAD) of the Silencing Mediator for Retinoid and Thyroid hormone receptors (SMRT): SMRT-HDAC3 corepressor complex, which is required for transcriptional repression [118,119].

HDACs do not act in isolation; they form complexes and coordinate activity, often working with corepressors and chromatin remodellers to regulate transcription [120]. Because of functional redundancy between HDACs, targeting a single HDAC is often insufficient to block epigenetic regulation, requiring the inhibition of multiple HDACs for a significant effect [121]. Targeting IPMK, rather than individual HDACs, could offer a broader disruption of epigenetic mechanisms with fewer compensatory effects, enhancing therapeutic efficacy.

Given that *P. falciparum* could have a similar mechanism in which IPMK-generated inositol phosphates regulate chromatin remodelling, we hypothesize that these polyphosphates could stabilize HDACII function. If substantiated, this could further influence transcriptional programs that control antigenic variation, gametocyte differentiation and gametogenesis. Notably, IP4 has been identified as a key regulator of microgametocyte exflagellation in *P. falciparum* since 1994 [122].

This suggests that targeting IPMK, which could indirectly modulate the activity of HDACs, could be a superior strategy for disrupting epigenetic regulation in *P. falciparum*. Moreover, given its connection to HDACII, IPMK may regulate transcription through chromatin remodelling, supporting its role in antigenic variation and gametocyte development.

### Potential roles in calcium signalling

The importance of calcium signalling in *P. falciparum* is well established [123,124]. *P. falciparum* IPMK may contribute to calcium homeostasis by shifting the utilization of IP3 towards the production of higher inositol polyphosphates, as observed in *C. elegans* [116]. At low cellular calcium levels, IP4 may be hydrolysed to IP3. Additionally, IP4 can prevent IP3 dephosphorylation under these conditions, thereby actively balancing intracellular calcium levels, as extensively reviewed [125].

### *P. falciparum* IPMK as a drug target

This study indicates that IPMK in the *P. falciparum* inositol pyrophosphate pathway may be essential for parasite survival across multiple life stages, although its contribution to the synthesis of higher levels of inositol pyrophosphate remains questionable. The essentiality of the multidomain gene containing IPMK has been determined through piggyBac mutagenesis studies and with a destabilization domain (DD) [63,74]. This essentiality is further emphasized by population-level conservation patterns.

Consistent with this, population-level sequence analyses in PlasmoDB [60] shows that PF3D7_1008000 is highly conserved in 366 *P. falciparum* isolates, of which 178 are non-synonymous and 195 are synonymous, giving a non-synonymous-to-synonymous ratio of 0.91 out of 486 catalogued SNPs. Only one stop-gain mutation was observed, which highlights the rarity of loss-of-function variants. Such widespread conservation across diverse parasite populations suggests that the protein is functionally indispensable and less prone to disruptive mutations. Given the persistent challenge of antimalarial drug resistance, including the well-documented tendency of kinases to acquire active-site mutations under

drug pressure, targeting structurally unique or allosteric regions of *P. falciparum* IPMK rather than the conserved catalytic site may reduce this risk and further support its prioritization as a drug target, particularly within combination therapies that are designed to address both efficacy and long-term resistance.

*P. falciparum* IPMK is a nuclear and cytosolic enzyme enriched in rings, trophozoites, schizonts, and stage IV and V gametocytes in humans and also present in oocysts and salivary gland sporozoites in mosquitoes [9,74,108–110,112]. Successful inhibitors of *P. falciparum* IPMK may therefore align with malaria target product profiles (TPPs) for case management and chemoprotection, satisfying all categories of Target Compound Profiles (TCPs), except for TCP4 (agents targeting liver schizonts) [126,127].

Bioinformatics plays a crucial role in identifying and characterizing drug targets, enhancing success rates and significantly reducing the cost of drug discovery [128]. This study provides the first sequence-level evidence that *P. falciparum* IPMK is conserved, non-redundant, structurally distinct, and paralogous to human IPMK, thereby representing a druggable and selectively targetable enzyme. Our results indicate the possibility of pan-species activity and highlight structural characteristics that can be investigated for selective inhibition.

## Conclusion

This study refines our understanding of *P. falciparum* IPKs by reclassifying the gene previously annotated as IPK2 in PlasmoDB as an atypical IP6K and identifying a conserved QxxxDxKxG substitution in IPMK1 that distinguishes it from canonical IPMKs. Structural modelling suggests that IPMK2 may be the only catalytically viable and selectively targetable member of this family in *P. falciparum*, pointing to its potential as a novel antimalarial drug target. While our study focused on sequence- and structure-based comparisons, future work should also include experimental analyses at the protein level. Particularly, direct comparison of *P. falciparum* and human IPMK proteins by western blot or related assays would provide valuable validation of parasite–host differences in expression and molecular features. Although experimental validation will be required to establish enzyme function and confirm druggability, including recombinant assays and compound library screening, these findings provide a strong foundation for future work and highlight IPMK2 as a promising candidate for selective therapeutic intervention in malaria.

## Supporting information

**S1 File.** S1 Fig. Multiple sequence alignment of IPMK proteins from *Plasmodium* species that infect humans. Sequences from *P. falciparum* (Q8I3W0), *P. malariae* (A0A1D3RII6), *P. vivax* (A0A564ZW73), *P. ovale* (A0A1D3U8P3) and *P. knowlesi* (B3L6B8) were aligned with Clustal Omega and viewed in Jalview. The sequences in the grey frame highlight the identified signature motifs QxxxDxKxG, SLL and IDF. Invariant residues are shaded blue. S1 Table. List of IPK proteins used for structural analysis and their annotations. S2 Fig. Modelling and structure assessment of *P. falciparum* IPMK1. (A & C) Local distance difference test (LDDT) score per position predicted for the five *P. falciparum* IPMK1_kinase domain models generated by Colabfold. The model predictions had high- and low-confidence regions, and the low-confidence areas (red, blue and orange frames) were intrinsically disordered regions. (B) PAE plot of the best model predicted, which was refined and analysed on ProSA-Web (E) and SAVES (D), showing that 94.5% was Ramachandran favoured. S3 Fig. 3D modelling of the PfIPMK kinase domain by SWISS-MODEL. (A) Template information. (B) Model assessment of the predicted *P. falciparum* IPMK2 domain showing a Ramachandra plot of the initial model structure and that of the refined model, which has 94.4% of its residues in the favoured regions (D). (C) ProSA-web output of the quality assessment of the refined model. S2 Table. Table of druggable pockets predicted by PockDrug for *P. falciparum* IPMK2 and their parameters. S3 Table. Summary of the predicted subcellular locations of the putative *P. falciparum IPKs.* S1 Sequence. Predicted amino acid sequence of P. falciparum 3D7 IPMK1. S4 Table. Peptides mapped to enzymes in the *P. falciparum* IPP. **S1 document.** Uncropped images.
(ZIP)

## Author contributions

**Conceptualization:** Abigail Obuobi, Neils B. Quashie, Nancy Odurowah Duah-Quashie, Jon R. Sayers.

**Formal analysis:** Abigail Obuobi, Jon R. Sayers.

**Funding acquisition:** Abigail Obuobi, Jon R. Sayers.

**Investigation:** Abigail Obuobi.

**Methodology:** Abigail Obuobi, Neils B. Quashie, Nancy Odurowah Duah-Quashie.

**Resources:** Neils B. Quashie, Nancy Odurowah Duah-Quashie.

**Supervision:** Neils B. Quashie, Nancy Odurowah Duah-Quashie, Jon R. Sayers.

**Validation:** Jon R. Sayers.

**Writing – original draft:** Abigail Obuobi.

**Writing – review & editing:** Abigail Obuobi, Neils B. Quashie, Nancy Odurowah Duah-Quashie, Jon R. Sayers.

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
