## [Decision Letter · Decision Letter 0]

5 May 2025

Dear Dr. Sayers,

Thank you for submitting your manuscript to PLOS ONE. After careful consideration, we feel that it has merit but does not fully meet PLOS ONE’s publication criteria as it currently stands. Therefore, we invite you to submit a revised version of the manuscript that addresses the points raised during the review process.

We look forward to receiving your revised manuscript.

Kind regards,

Yash Gupta, Ph.D.

Academic Editor

PLOS ONE

3. Please include a complete copy of PLOS’ questionnaire on inclusivity in global research in your revised manuscript. Our policy for research in this area aims to improve transparency in the reporting of research performed outside of researchers’ own country or community. The policy applies to researchers who have travelled to a different country to conduct research, research with Indigenous populations or their lands, and research on cultural artefacts. The questionnaire can also be requested at the journal’s discretion for any other submissions, even if these conditions are not met.  Please find more information on the policy and a link to download a blank copy of the questionnaire here: https://journals.plos.org/plosone/s/best-practices-in-research-reporting. Please upload a completed version of your questionnaire as Supporting Information when you resubmit your manuscript.

4. Please remove your figures from within your manuscript file, leaving only the individual TIFF/EPS image files, uploaded separately. These will be automatically included in the reviewers’ PDF.

6. We notice that your supplementary figures are uploaded with the file type 'Figure'. Please amend the file type to 'Supporting Information'. Please ensure that each Supporting Information file has a legend listed in the manuscript after the references list.

Additional Editor Comments:

Authors need to systematically address reviewers comments. If extra time is required for incorporating all please let us know.

Reviewers' comments:

Reviewer's Responses to Questions

**Comments to the Author**

1. Is the manuscript technically sound, and do the data support the conclusions?

Reviewer #1: No

Reviewer #2: Partly

Reviewer #3: Partly

2. Has the statistical analysis been performed appropriately and rigorously?

Reviewer #1: N/A

Reviewer #2: N/A

Reviewer #3: N/A

3. Have the authors made all data underlying the findings in their manuscript fully available?

Reviewer #1: Yes

Reviewer #2: Yes

Reviewer #3: Yes

4. Is the manuscript presented in an intelligible fashion and written in standard English?

Reviewer #1: Yes

Reviewer #2: Yes

Reviewer #3: Yes

Reviewer #1: In the manuscript titled 'Unveiling the Plasmodium inositol (pyro)phosphate pathway: highlighting inositol polyphosphate multikinase as a novel therapeutic target for malaria', the authors used computational approaches to annotate IPKs, with a special focus on IPMK, within the context of the inositol phosphate pathway. They present IPMK as a promising therapeutic target for P. falciparum malaria. The manuscript includes a comparative analysis of four selected PfIPKs against their human counterparts and homologues from other organisms. While the study adds to understand the molecular landscape of the IPP pathway, there are many major concerns and points need attention and clarification. While the authors have made commendable efforts in analysing the tertiary structures of the proteins, there remains considerable scope for improving the interpretation and integration of these structural insights into the overall findings. Comments are attached in a separate file.

Reviewer #2: Abigail Obuobi et al. in their manuscript has described the importance of inositol phosphate multi-kinases (IPMK) as attractive drug targets against malaria. The authors have talked about the durability of IPMK and the selectivity the drugs will have due to their differences in structure from the human counterpart.

The plasmodium IPMK has a novel variation as described in result “Identification of conserved motifs and residues” which could potentially alter the structure of the protein.

Structural analysis also suggests these proteins are different from the human counterpart making them an attractive target in drug discovery against malaria.

The work however lacks conviction due to the lack of in-vitro work supporting the in-silico part which is only limited genomic DNA PCR to confirm the presence of the IPMK in the genome of plasmodium.

More in-vitro work needs to be done to check the expression levels and pattern of these proteins. (Western blot if antibodies are available or human antibodies can detect these proteins)

Real time PCR to check or the transcription of these genes and role in life cycle stages etc.

Screen compound libraries like the inositol phosphate libraries on plasmodium blood culture and human cell lines to look for inhibition and selectivity of already known inhibitors of these proteins.

Reviewer #3: Drug-resistant malaria necessitates the discovery of new drugs with novel mechanisms. The inositol phosphate signaling pathway (IPP) in Plasmodium falciparum is a promising target due to its crucial roles in parasite metabolism and signaling. Researchers investigated this pathway using computational and molecular methods. They characterized the parasite's inositol phosphate kinases in silico and confirmed the presence of their genes. The study provides molecular evidence for the IPP's existence in P. falciparum and suggests that targeting this pathway could be lethal to the parasite. Specifically, P. falciparum inositol polyphosphate multikinase (IPMK) is highlighted as a particularly attractive drug target because of its unique features.

Comments:

1) The article is well written and have no issue in English writing

2) In the Figure 13 Gel image shows all the presence of the p falciparum IPMK1. It would have been more informative if the same gene from human would been shown in the western blot

3) The second major concern is from Figure 3. Amino acid alignment shows conserved regions from P falciparum and Human. These seems to share the conserved regions. Are these conserved regions part of active sites or not. if yes, authors have suggested that it can be targeted in Plasmodium falciparum which seems to be skeptical.

**Do you want your identity to be public for this peer review?** For information about this choice, including consent withdrawal, please see our Privacy Policy

Reviewer #1: No

Reviewer #2: No

Reviewer #3: No

---

## [Author Response · Author response to Decision Letter 1]

15 Oct 2025

I hope we have responded to all the Office comments about supporting information etc and additional documents as requested including responses to reviewer comments.

All office queries/requestys have been responded to in the submitted documents to the best of our ability/knowledge.

Response to Reviewers

Reviewer 1

1. While the authors have made commendable efforts in analysing the tertiary structures of the proteins, there remains considerable scope for improving the interpretation and integration of these structural insights into the overall findings. It would be interesting if the authors correlated the active pocket residues and spatial location of the motif, as well as the regions contributing to the lower RMSD values, which might have further strengthened the hypothesis.

Response:

We have expanded the structural analysis to directly integrate active-site context and motif positioning into our interpretation. Specifically, in the revised version we have correlated the active-site residues with the spatial location of the conserved motifs and have highlighted how the QxxxDxKxG substitution in P. falciparum IPMK1 may influence binding interactions. To illustrate these relationships, we have added a 2D interaction map and extended our discussion of regions contributing to the lower RMSD values in the superimpositions. These additions (Results, pp. 26–29; lines 627–697, Fig. 11) provide a clearer mechanistic link between sequence divergence and structural/functional implications.

2. Most of the analyses presented in the study are computational. Although the authors amplified and sequenced the IPK genes, this adds little new information apart from confirming a previously known change in the IPMK1 motif. In several instances, the conclusions drawn either lack strong supporting evidence or restate already known facts

Response:

We appreciate this comment and understand the concern. However, we would like to respectfully clarify that our study contributes new insights beyond restating existing knowledge. For instance, while P. falciparum IPMK1 has long been annotated in PlasmoDB, it has not previously been subjected to detailed structural or motif-level analyses, likely because earlier reports may have assumed it lacked a complete catalytic signature. Here, we demonstrated for the first time that IPMK1 carries a QxxxDxKxG catalytic motif substitution. This was validated through sequencing in a field isolate and confirmed in in silico to be conserved across human-infecting Plasmodium species. We have revised the “Identification of Conserved Motifs and Residues” section to reflect this more clearly (pp. 15–16; lines 360–374).

In addition, P. falciparum IPMK2, which is embedded within a multidomain gene annotated as HDAC2, has not been delineated or analysed as a distinct kinase before. Our analysis clarifies its identity and evolutionary placement within the Plasmodium IPMK clade.

Finally, our in silico analysis provides the first systematic evidence that the gene previously annotated in PlasmoDB as “IPK2” is in fact an atypical IP6K, thereby refining its evolutionary classification and revealing an unusual similarity to human IPMK (p. 19; lines 469-471).

Taken together, these findings establish a more accurate picture of the P. falciparum IPK family at the domain, phylogenetic, and structural levels. We believe these clarifications extend beyond confirmation, providing a foundation for future functional validation and for exploring these enzymes as potential drug targets.

3. The authors state that 'the presence of their genes was confirmed via conventional PCR and sequencing'… confirming the mere presence of these genes seems unnecessary, as their existence in the genome is already well established. Such an approach would only be justified if the study involved highly divergent isolates or aimed to detect strain-specific variation. The authors should revise the statement to clarify that the objective was to validate the mutation, not the gene’s presence — or provide a clear rationale.

Response:

We agree that the existence of these genes in the P. falciparum genome is already well established in PlasmoDB and other reference databases. Our objective was not to reconfirm their presence per se, but rather to assess sequence integrity in both laboratory strains and a clinical isolate, with a particular emphasis on experimentally verifying the p.Pro375Gln substitution in IPMK1, which, to our knowledge, had not previously been confirmed in field isolates. To make this clearer and in line with your suggestion, we have revised the text to highlight that our PCR and sequencing experiments were performed to validate the mutation and confirm sequence integrity, rather than to demonstrate gene presence. These clarifications have been incorporated into the revised manuscript (see Confirmation of the sequences of P. falciparum IPKs section; p. 31, lines 746–764).

4. In the section 'Phylogenetic analysis and comparative genomics', the authors compare the kinase protein of P. falciparum with its human homologues using both BLASTp and pairwise alignment. However, it is unclear whether the full-length protein or only the kinase domain was used in these analyses. If only the domain was considered, the authors should explain the notable differences in percentage similarity observed between the two methods, particularly in the case of the VIP1 gene. While it is understood that BLASTp performs local alignment and pairwise alignment can be either local or global, the discrepancy in similarity results is difficult to reconcile without clarification. These two paragraphs require revision to ensure consistency and to support a unified conclusion.

Response:

We confirm that, except for the initial domain searches, all comparative and phylogenetic analyses were performed using kinase domain sequences. This includes both BLASTp searches and pairwise alignments. The apparent discrepancies in percent similarity arise from the different alignment strategies: BLASTp reports local similarity within the best-matching region, while Jalview calculates global identity across the full kinase domain. For P. falciparum VIP1, both methods consistently identified H. sapiens PPIP5K1 as the closest homologue, and the reported similarities (44.91% vs. 45%) are effectively identical, confirming the consistency of the approaches. The larger differences observed for P. falciparum IPMK2 and IPK2 reflect the fact that BLASTp retrieved H. sapiens IP6K3 as the closest local match, whereas pairwise alignment compared the parasite sequences directly with H. sapiens IPMK and H. sapiens IP6K1. We have clarified this explicitly in the Methods (p. 6; lines 135–145) and Results (pp. 17–19; lines 406-407, 439, 456-463) sections of the revised manuscript.

5. The authors first state that the QxxxDxKxG motif is specific to P. falciparum, but in the following paragraph, they report its presence in all Plasmodial IPMK1 sequences. This appears contradictory and should be clarified.

Response:

Our intention was not to imply that the QxxxDxKxG motif is unique only to P. falciparum, but rather that it represents a conserved distinguishing feature of Plasmodial IPMK1 proteins more broadly. To clarify, we first identified the substitution in P. falciparum IPMK1 and then extended our analysis across other human-infecting Plasmodium species, where the same motif was also present. We now state explicitly that the QxxxDxKxG motif is a conserved characteristic of the Plasmodium IPMK1 clade in the revised version. Corresponding revisions are included in the “Identification of conserved motifs and residues” section (pp 15-16; lines 357–374).

6. When P. falciparum IP6K is compared with H. sapiens IPK enzymes, it is more closely related to H. sapiens IP6Ks than to H. sapiens IPMK, as indicated by the percentage sequence identity above.” Isn’t it obvious that P. falciparum IP6K will be more similar to human IP6Ks than to IPMK. Unless this comparison addresses a specific ambiguity or unexpected result, it may not be necessary. The authors could clarify the motivation behind highlighting this point.

Response:

We agree that the earlier wording could make the comparison seem self-evident. Our intention, however, was to highlight an atypical relationship: although identified as an IP6K, P. falciparum IP6K shows an unusual degree of similarity to H. sapiens IPMK, in some comparisons exceeding its similarity to H. sapiens IP6Ks. We have revised the text to clarify this point, emphasizing that the motivation for the comparison was to draw attention to this unexpected evolutionary signal rather than to restate the obvious. (p. 19-20, lines 469–474).

7. The paragraph (lines 463–467) refers to Figure 6 to demonstrate the syntenic nature of IPMK1; however, the figure appears to depict a phylogenetic tree of the VIP1 gene instead. The authors should clarify this discrepancy and ensure that the correct figure is cited.

Response

We agree that the original citation of Fig. 6 was indeed incorrect, as that figure depicts a phylogenetic tree of the VIP1 gene rather than synteny analysis. In the revised version, we have corrected the text: the reference to Fig. 6 has been removed, and we now clarify that the synteny observation is based on sequence analysis (data not shown). The revised text appears in the Results section (pp. 20-21, lines 496–498).

8. The authors state that P. falciparum IPMK1 is “evolutionarily older.” In Figure 5, what do the branch lengths represent? If they are substitutions per site, this reflects evolutionary change, not time, unless a molecular clock is used. Moreover, earlier sections describe motif divergence, which is more consistent with functional divergence over time than with evolutionary age. Could the authors clarify the basis for this statement and explain how they infer evolutionary age in the absence of a time-calibrated phylogeny?

Response:

We agree that our original phrasing “evolutionarily older” was imprecise, since our phylogeny was not time-calibrated. We have revised the text to state that P. falciparum IPMK1 “occupies a more basal position within the Plasmodium IPMK clade, indicating greater phylogenetic divergence from other eukaryotic IPMKs.” We further clarify that this divergence is accompanied by the QxxxDxKxG catalytic motif substitution, which may impair function, in contrast to P. falciparum IPMK2, which retains the canonical motif. The revised text avoids implying evolutionary timing and instead highlights relative phylogenetic and functional divergence (pp.20 , lines 477–482).

Reviewer 2

1. The reviewer notes that the manuscript lacks in vitro validation and suggests additional experiments such as Western blot, qPCR, and compound library screening to confirm expression and druggability.

Response:

We thank the reviewer for this valuable suggestion and fully agree that experimental validation will be essential to confirm the biological roles and druggability of P. falciparum IPMKs. While our current study did not include assays such as RT-PCR, western blot, or compound library screening, we sought to strengthened the manuscript by incorporating stage-specific expression data from mass spectrometry–based proteomic and RNA-seq datasets available in PlasmoDB (see revised Methods, “Subcellular localization, functional and analyses.,” p.10; lines 228–231; Results, pp. 32; lines 776-790). We have taken an additional step to reorganize this section into focused sub-sections (metabolic, nuclear/epigenetic, and calcium-related roles), which make the potential functional contributions of IPMKs across parasite stages clearer (pp.32-34; lines 766, 792, 802, 815, 838). Finally, the Conclusion has been revised to explicitly acknowledge the need for in vitro and chemical validation, while emphasizing that our findings provide a strong framework for prioritizing and guiding such future; experimental work. (See Conclusion p. 36; lines 878-890). We will definitely consider these suggested experiments in future work.

2. Screen compound libraries like the inositol phosphate libraries on plasmodium blood culture and human cell lines to look for inhibition and selectivity of already known inhibitors of these proteins.

Response:

We agree that compound screening, including inositol phosphate inhibitor libraries, will be an essential next step to assess druggability. However, as is well established for kinases, inhibitor promiscuity poses a major challenge, and whole-parasite screening alone cannot resolve target specificity. Confirming P. falciparum IPMK2 (and other IPKs) as bona fide molecular targets will require target deconvolution approaches or, preferably, recombinant expression and biochemical assays prior to whole-parasite validation. To reflect this, we have revised the Conclusion to emphasize that future work should integrate biochemical, cellular, and chemical validation, including compound library screening, to establish P. falciparum IPMK2 as a selective and druggable target (p. 36, lines 886-890).

Reviewer 3

1. The article is well written and have no issue in English writing

Response:

We thank the reviewer for their positive assessment and for confirming that the manuscript is clearly written in English. We appreciate their encouraging feedback.

2. In the Figure 13 Gel image shows all the presence of the p falciparum IPMK1. It would have been more informative if the same gene from human would been shown in the western blot

Response:

We would like to clarify that Figure 13 (now Figure 14) presents PCR amplification products resolved on agarose gel, rather than a western blot. In this study, our focus was on verifying the presence and sequence integrity of P. falciparum IPMK1 (including the p.Pro375Gln substitution), rather than comparative protein-level expression. We fully agree that incorporating human IPMK protein as a control via western blot would provide valuable complementary insights. Although this was beyond the scope of the present work, we have now noted in the revised Conclusion that a direct experimental comparison with human IPMK will be an important future direction for validating parasite–host differences. (p.36, lines 884–886).

3. The second major concern is from Figure 3. Amino acid alignment shows conserved regions from P. falciparum and Human. These seems to share the conserved regions. Are these conserved regions part of active sites or not. if yes, authors have suggested that it can be targeted in Plasmodium falciparum which seems to be sceptical.

Response:

Several conserved residues identified in our alignment do fall within the canonical ATP-binding region. However, P. falciparum IPMK1 appears catalytically impaired due to the QxxxDxKxG substitution and is therefore not considered a viable target. By contrast, P. falciparum IPMK2 retains a functional ATP-binding geometry while also carrying parasite-specific insertions and structural differences from the human enzyme. These changes reshape the local pocket environment and create additional druggable cavities (Fig. 12). Supporting this, population-level data from PlasmoDB show that PF3D7_1008000, which encodes IPMK2 within a multidomain gene, is highly conserved across 366 isolates, with only one stop-gain mutation reported (p. 35; lines 853-864). Such conservation points to its functional indispensability but also suggests that targeting parasite-specific or allosteric regions, rather than the conserved catalytic core, may help mitigate resistance. The Results and Discussion has been revised to make clear that our selectivity arguments apply specifically to P. falciparum IPMK2 and not to conserved hinge residues (p. 27, lines 649–654).

---

## [Decision Letter · Decision Letter 1]

24 Nov 2025

Unveiling the Plasmodium inositol (pyro)phosphate pathway: highlighting inositol polyphosphate multikinase as a novel therapeutic target for malaria

PONE-D-25-19273R1

Dear Dr. Sayers,

We’re pleased to inform you that your manuscript has been judged scientifically suitable for publication and will be formally accepted for publication once it meets all outstanding technical requirements.

Kind regards,

Yash Gupta, Ph.D.

Academic Editor

PLOS ONE

Additional Editor Comments (optional):

Reviewers' comments:

Reviewer's Responses to Questions

**Comments to the Author**

Reviewer #1: All comments have been addressed

Reviewer #3: All comments have been addressed

2. Is the manuscript technically sound, and do the data support the conclusions?

Reviewer #1: Yes

Reviewer #3: Yes

3. Has the statistical analysis been performed appropriately and rigorously?

Reviewer #1: N/A

Reviewer #3: Yes

4. Have the authors made all data underlying the findings in their manuscript fully available?

Reviewer #1: Yes

Reviewer #3: Yes

5. Is the manuscript presented in an intelligible fashion and written in standard English?

Reviewer #1: Yes

Reviewer #3: Yes

Reviewer #1: I appreciate the authors’ responses to the concerns raised. The revisions in the current manuscript have clarified the earlier points and improved the overall quality. As discussed in the comments, and as the authors agreed, “Our objective was not to reconfirm their presence per se, but rather to assess sequence integrity in both laboratory strains and a clinical isolate, with a particular emphasis on experimentally verifying the p.Pro375Gln substitution in IPMK1, which, to our knowledge, had not previously been confirmed in field isolates”. This is the important and key point, which should be highlighted in the abstract. In line with this, authors could consider updating the sentence in the abstract (line 26): rather than stating that the presence of the gene was confirmed, it would be clearer to say that the gene sequence or locus was validated in the strains, with emphasis on the experimental verification of the mentioned substitution. With these updates, I believe the manuscript is now suitable for acceptance, and I congratulate the authors on their work.

Reviewer #3: All comments have been addressed and aligns with the claim of the authors. The article will add to the knowledge of the plasmodium therapeutic target.

**Do you want your identity to be public for this peer review?** For information about this choice, including consent withdrawal, please see our Privacy Policy

Reviewer #1: No

Reviewer #3: No

---

## [Editor Report · Acceptance letter]

1 Dec 2025

PONE-D-25-19273R1

PLOS ONE

Dear Dr. Sayers,

I'm pleased to inform you that your manuscript has been deemed suitable for publication in PLOS ONE. Congratulations! Your manuscript is now being handed over to our production team.

Kind regards,

on behalf of

Dr. Yash Gupta

Academic Editor

PLOS ONE